# THE ART OF SCALING REINFORCEMENT LEARNING COMPUTE FOR LLMS

**Devvrit Khatri**[*][△◇]  **Lovish Madaan**[*][†◇]  **Rishabh Tiwari**[‡◇]  **Rachit Bansal**[Γ◇]  **Sai Surya Duvvuri**[△◇]
**Manzil Zaheer**[◇]  **Inderjit S. Dhillon**[△Υ]  **David Brandfonbrener**[◇]  **Rishabh Agarwal**[◇]
[△]UT Austin  [◇]Meta  [†]UCL  [‡]UCB  [Γ]Harvard University  [Υ]Google

## ABSTRACT

Reinforcement learning (RL) has become central to training large language models (LLMs), yet the field lacks predictive scaling methodologies comparable to those established for pre-training. Despite rapidly rising compute budgets, there is no principled understanding of how to evaluate algorithmic improvements for scaling RL compute. We present the first large-scale systematic study, amounting to more than 400,000 GPU-hours, that defines a principled framework for analyzing and predicting RL scaling in LLMs. We fit sigmoidal compute-performance curves for RL training and ablate a wide range of common design choices to analyze their effects on asymptotic performance and compute efficiency. We observe: (1) Not all recipes yield similar asymptotic performance, (2) Details such as loss aggregation, normalization, curriculum, and off-policy algorithm primarily modulate compute efficiency without materially shifting the asymptote, and (3) Stable, scalable recipes follow predictable scaling trajectories, enabling extrapolation from smaller-scale runs. Combining these insights, we propose a *best-practice* recipe, **SCALERL**, and demonstrate its effectiveness by successfully scaling and predicting validation performance on a single RL run scaled up to 100,000 GPU-hours. Our work provides both a *scientific framework* for analyzing scaling in RL and a practical recipe that brings RL training closer to the predictability long achieved in pre-training.

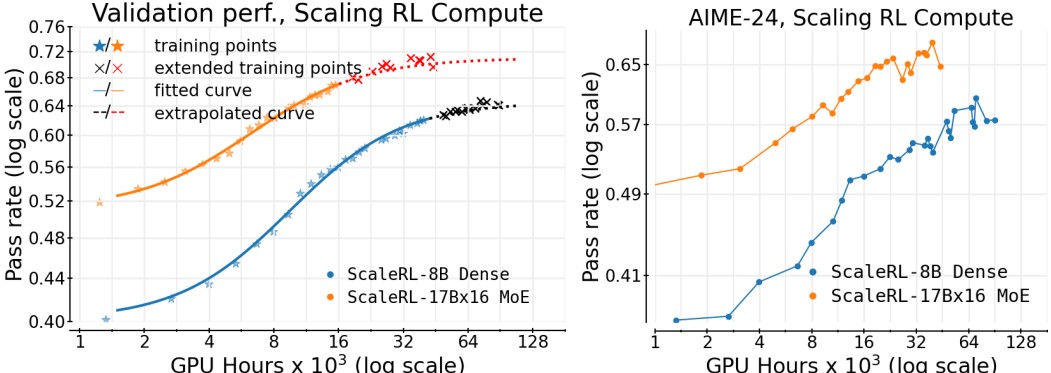

Figure 1: **Predicatably Scaling RL compute to 100,000 GPU Hours** (a) We run **SCALERL** for 100k GPU hours on an 8B dense model, and 50k GPU hours on a 17Bx16 MoE (Scout). We fit a sigmoid curve (Equation (1)) on pass rate (mean@16) on *iid* validation dataset up to 50k (and 16k) GPU hours and extrapolate to 100k (and 45k) on the 8B (Scout MoE) models respectively. We trained for 7400 steps for 8B and 7100 steps for Scout, which is $3.5\times$ larger than ProRL (Liu et al., 2025a). The extrapolated curve ($\times$ markers) closely follows extended training, demonstrating both stability at large compute and predictive fits–establishing **SCALERL** as a reliable candidate for RL scaling. (b) **Downstream evaluation on AIME-24** shows a consistent scaling trend for **SCALERL**, thus generalizing beyond the training data distribution. Moreover, scaling model size substantially improves the downstream and asymptotic RL performance.

---

[*]Equal technical contribution

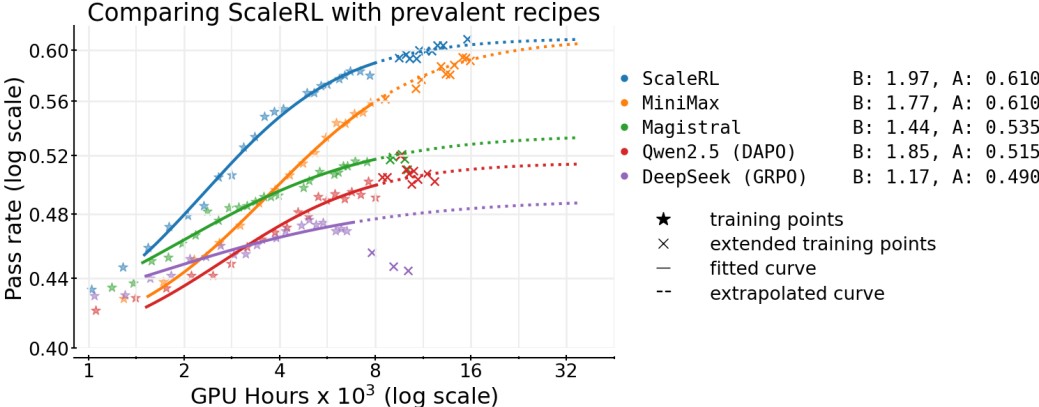

Figure 2: **SCALERL is more scalable than prevalent RL methods**. We fit sigmoid curves (Equation 1) on *iid* validation dataset to commonly-used training recipes like DeepSeek (GRPO) (Guo et al., 2025), Qwen-2.5 (DAPO) (Yu et al., 2025), Magistral (Rastogi et al., 2025), and Minimax-M1 (MiniMax et al., 2025), and compare them with **SCALERL**. **SCALERL** surpasses all other methods, achieving an asymptotic reward of $A = 0.61$. Stars denote evaluation points; solid curves show the fitted curve over the range used for fitting; dashed curves extrapolate beyond it. We validate the predictability by running each method for longer ("$\times$" markers), which align closely with the extrapolated curves for stable recipes like **SCALERL** and MiniMax. Further description of the individual recipes compared are given in Appendix A.17.

## 1  INTRODUCTION

Scaling reinforcement learning (RL) compute is emerging as a critical paradigm for advancing large language models (LLMs). While pre-training establishes the foundations of a model; the subsequent phase of RL training unlocks many of today's most important LLM capabilities, from test-time thinking (OpenAI, 2024; Guo et al., 2025) to agentic capabilities (Kimi Team et al., 2025a). For instance, Deepseek-R1-Zero used 100,000 H800 GPU hours for RL training – 3.75% of its pre-training compute (Guo et al., 2025). This dramatic increase in RL compute is amplified across frontier LLM generations, with more than $10\times$ increase from o1 to o3 (OpenAI, 2025) and a similar leap from Grok-3 to Grok-4 (xAI Team, 2025).

While RL compute for LLMs has scaled massively, our understanding of *how* to scale RL has not kept pace; the methodology remains more *art* than science. Recent breakthroughs in RL are largely driven by isolated studies on novel algorithms (e.g., Yu et al. (DAPO, 2025)) and model-specific training reports, such as, MiniMax et al. (2025) and Magistral (Rastogi et al., 2025). Critically, these studies provide *ad-hoc* solutions tailored to specific contexts, but not *how* to develop RL methods that scale with compute. This lack of scaling methodology stifles research progress: with no reliable way to identify promising RL candidates *a priori*, progress is tied to large-scale experimentation that sidelines most of the academic community.

This work lays the groundwork for science of RL scaling by borrowing from the well-established concept of *scaling laws* from pre-training. While pre-training has converged to algorithmic recipes that scale *predictably* with compute (Kaplan et al., 2020; Hoffmann et al., 2022; Owen, 2024), the RL landscape lacks a clear standard. As a result, RL practitioners face an overwhelming array of design choices, leaving the fundamental questions of *how* to scale and *what* to scale unanswered. To address these, we establish a predictive framework for RL performance using a sigmoid-like saturating curve between the expected reward ($R_C$) on an *iid* validation set and training compute ($C$):

$$\underbrace{R_C - R_0}_{\text{Reward Gain}} = \underbrace{(A - R_0)}_{\text{Asymptotic Reward Gain}} \times \underbrace{\frac{1}{1 + (C_{\text{mid}}/C)^B}}_{\text{Compute Efficiency}} \qquad \textit{(fixed model and training data)} \qquad (1)$$

where $0 \leq A \leq 1$ represents the asymptotic pass rate, $R_0$ represents initial reward at 0 training compute, $B > 0$ is a scaling exponent that determines the compute efficiency, and $C_{\text{mid}}$ sets the midpoint of the RL performance curve. A schematic interpretation of these parameters is provided in Figure 3.

This framework in Equation (1) allows researchers to extrapolate performance from lower-compute runs to higher compute budgets, enabling them to evaluate scalability of RL methods without incurring the compute cost of running every experiment to its computational limit.

Guided by this framework, we develop **SCALERL**, an RL recipe that scales *predictably* with compute. In a massive **100,000 GPU-hours training run**, we show that **SCALERL**'s performance closely matches the scaling curve predicted by our framework (Figure 1). Critically, scaling curves extrapolated from only the initial stages of training closely match the final observed performance, confirming the predictive ability of our framework to extreme compute scales.

The design of **SCALERL** is grounded in a comprehensive empirical study of RL scaling that spanned over **400,000 GPU-hours** (on Nvidia GB200 GPUs). This study explored numerous design choices at an 8B model parameters scale, where individual runs use up to 16,000 GPU-hours, making them **6× cheaper** than experimenting at our largest training run scale. This investigation yielded three key principles:

- **RL Performance Ceilings are Not Universal**: As we scale training compute for different methods, they encounter different ceilings on their achievable performance ($A$). This limit can be shifted by choices such as the loss type and batch size.
- **Embracing the Bitter Lesson**: Methods that appear superior at small compute budgets can be worse when extrapolated to large-compute regimes (Figure 2). We can still identify scalable methods by estimating the scaling parameters ($A$, $B$) from the early training dynamics using our framework (Equation (1)).
- **Re-evaluating Common Wisdom**: Common interventions thought to improve peak performance (e.g., loss aggregation, data curriculum, length penalty, advantage normalization) mainly adjust compute efficiency ($B$), while not changing the performance ceiling considerably.

Based on these insights, **SCALERL** achieves *predictable* scaling by integrating existing methods, rather than inventing novel methods. Specifically, **SCALERL** combines asynchronous Pipeline-RL setup (§3.1), forced length interruptions, truncated importance sampling RL loss (CISPO), prompt-level loss averaging, batch-level advantage normalization, FP32 precision at logits, zero-variance filtering, and No-Positive-Resampling – with each component's contribution validated in a leave-one-out ablation, consuming 16,000 GPU-hours per run.

**SCALERL** not only scales *predictably* but also establishes a new **state-of-the-art** (Figure 2) – it achieves higher asymptotic performance and compute efficiency compared to established RL recipes. Moreover, **SCALERL** maintains predictable scaling when increasing compute across multiple training axes (§ 5) – including 2.5× larger batch sizes, longer generation lengths up to 32,768 tokens, multi-task RL using math and code, and larger MoE (Llama-4 17B×16); with benefits that consistently transfer to downstream tasks. Overall, this work establishes a rigorous methodology for cost-effectively predicting the scalability of new RL algorithms.

## 2 PRELIMINARIES & SETUP

We consider reinforcement learning with LLMs, where prompts $x$ are sampled from a data distribution $D$. Our setup follows a generator–trainer split across GPUs: a subset of GPUs (*generators*) use optimized inference kernels for high-throughput rollout generation, while the remaining GPUs (*trainers*) run the training backend (FSDP) and update parameters. We denote by $\pi_{\text{gen}}^{\theta}$ and $\pi_{\text{train}}^{\theta}$ the model with parameters $\theta$ on the generator and training backends, respectively. For each prompt, the old policy $\pi_{gen}^{\theta_{old}}$ on the generator GPUs produces candidate completions, which are then assigned scalar rewards. Policy optimization proceeds by maximizing a clipped surrogate objective, taking expectations over $x \sim D$ and rollouts from $\pi_{gen}^{\theta_{old}}$.

**Training Regimen** We mainly conduct our RL experiments using an 8B dense model on verifiable math tasks with a batch size of 768 and a maximum output sequence length of 14,336 tokens. More details about training, including SFT and hyper-parameters, are in Appendix A.3.

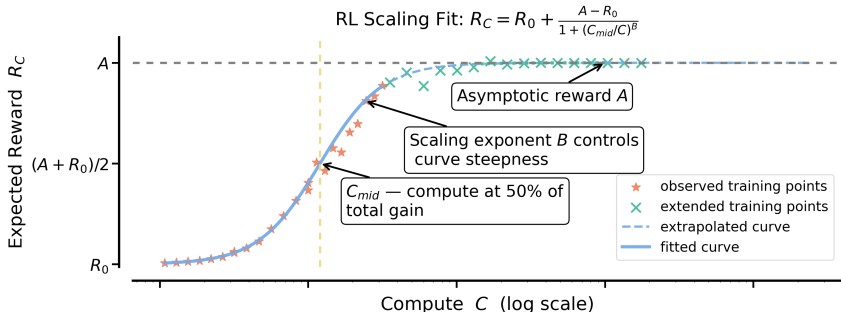

Figure 3: **Interpreting eq. (1)**. We provide an example fit illustrating the roles of parameters $A$, $B$, and $C_{\text{mid}}$. $C_{\text{mid}}$ determines the compute point at which half of the total gain is achieved - smaller values correspond to faster ascent toward the asymptote. $B$ controls the curve's steepness, with larger values indicating greater efficiency. $A$ represents the asymptotic performance reached at large compute scales. Further discussion is provided in Appendix A.8.

**Base RL Algorithm** As our starting point in § 3, we start with a "base" algorithm that resembles GRPO (Shao et al., 2024) without any KL regularization term, in line with large-scale training reports (Rastogi et al., 2025; MiniMax et al., 2025). Additionally, we include the asymmetric DAPO clipping (Yu et al., 2025), because of its widespread adoption as a default approach to avoid entropy collapse and maintain output diversity.

For a given prompt $x$, the old policy $\pi_{\text{gen}}(\theta_{\text{old}})$ generates $G$ candidate completions $\{y_i\}_{i=1}^{G}$, each assigned a scalar reward $r_i$. We compute advantages $\hat{A}_i$ and group-normalized advantages using:

$$\hat{A}_i = r_i - \text{mean}(\{r_j\}_{j=1}^{G}), \quad \hat{A}_i^G = \hat{A}_i/(\text{std}(\{r_j\}_{j=1}^{G}) + \epsilon).$$

Each completion $y_i$ of length $|y_i|$ contributes at the token-level importance sampling (IS) ratios $\rho_{i,t}(\theta)$, with asymmetric upper and lower clipping thresholds, akin to DAPO (Yu et al., 2025):

$$\rho_{i,t}(\theta) := \frac{\pi_{train}^{\theta}(y_{i,t} \mid x, y_{i,<t})}{\pi_{gen}^{\theta_{\text{old}}}(y_{i,t} \mid x, y_{i,<t})} = \frac{\pi_{train}^{\theta}(y_{i,t})}{\pi_{gen}^{\theta_{\text{old}}}(y_{i,t})}; \quad \text{clip}_{\text{asym}}(\rho, \epsilon^-, \epsilon^+) := \text{clip}(\rho, 1-\epsilon^-, 1+\epsilon^+).$$
$$(2)$$

We aggregate losses at the *sample level*, i.e., averaging per-sample token losses before averaging across samples. The surrogate objective is given by:

$$\mathcal{J}(\theta) = \mathbb{E}_{\substack{x \sim D, \\ \{y_i\}_{i=1}^{G} \sim \pi_{\text{gen}}^{\theta_{\text{old}}}(\cdot|x)}} \left[ \frac{1}{G} \sum_{i=1}^{G} \frac{1}{|y_i|} \sum_{t=1}^{|y_i|} \min\left( \rho_{i,t}(\theta)\hat{A}_i^G, \ \text{clip}_{\text{asym}}(\rho_{i,t}(\theta), \epsilon^-, \epsilon^+)\hat{A}_i^G \right) \right] \quad (3)$$

**Controlling Generation Lengths** To prevent reasoning output lengths from exploding during training, which harms training stability and efficiency, we use **interruptions** (GLM-V Team et al., 2025; Yang et al., 2025; Xu et al., 2025) that forcibly stop overly long generations by appending an end-of-thinking phrase (e.g., " `</think>` "), signaling the LLM to terminate its reasoning and produce a final answer. We revisit this choice later in Section 4 and compare it with length-penalty that penalizes long generations (Yu et al., 2025; Kimi Team et al., 2025b).

### 2.1 PREDICTIVE COMPUTE-SCALING AND FITTING CURVES

Unlike pre-training, which typically uses power-law to fit predictive curves, we model pass rate versus $\log(\textit{compute})$ with a sigmoidal function (Equation (1)). We do so because we found the sigmoidal fit to be much more robust and stable compared to power law empirically, which we discuss further in Appendix A.4. Moreover, our choice is consistent with prior work that use sigmoid-like power laws to capture bounded metrics such as accuracy (Ruan et al., 2024; Srivastava et al., 2022).

Similar to pre-training studies (Li et al., 2025b; Porian et al., 2025), we find that excluding the very early low-compute regime yields more stable fits, after which training follows a predictable

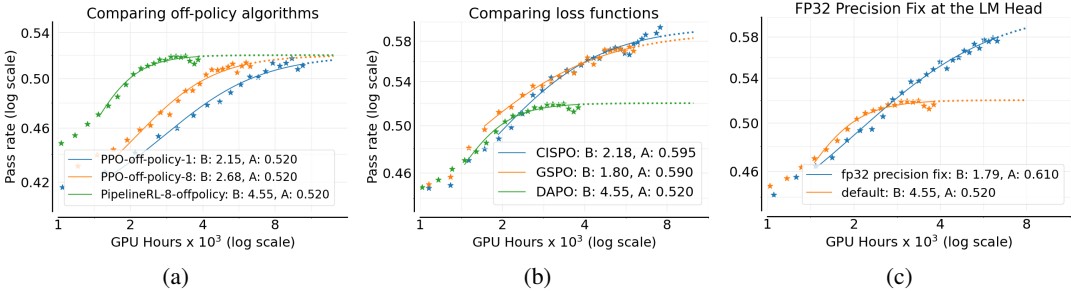

Figure 4: (a) **Comparing "compute-scaling" of asynchronous off-policy RL setups**. We report only the $B$ (scaling exponent) and $A$ (asymptotic pass rate) parameters of the fitted sigmoid curve (Equation 1). PipelineRL-$k$ is much more efficient and slightly better in the large compute limit. (b) **Comparing loss functions**: DAPO (Yu et al., 2025), GSPO (Zheng et al., 2025a), and CISPO (MiniMax et al., 2025). We find CISPO/GSPO achieve a higher asymptotic reward compared to DAPO. (b) **Using FP32 precision** in the final layer (LM head) gives a considerable boost in the asymptotic reward.

trajectory. Unless noted otherwise, all our scaling fits begin after ~1.5k GPU hours. Further details of the fitting procedure are provided in Appendix A.5 and the robustness of our curve fitting is discussed in Appendix A.7.

**Interpreting scaling curves**   Intuitively, a sigmoidal curve captures saturating returns - grows slowly in the low-compute regime, accelerates sharply through a mid-range of efficient scaling, and then saturates at high compute. We also provide a schematic interpretation of the parameters $A$, $B$, and $C_{mid}$ of the sigmoidal curve in Figure 3. We see that $B, C_{\text{mid}}$ primarily affects the efficiency of the run, and $A$ denotes the asymptotic performance at large compute scale. Further discussion of these parameters is provided in Appendix A.8.

**Scaling curve on held-out validation**   Consistent with pre-training practice (Hoffmann et al., 2022; Porian et al., 2025), we measure predictive performance on **in-distribution** validation data. Since our training runs span multiple epochs, we hold out randomly selected $1,000$ prompts from the Polaris-53k dataset for validation and use the remainder for training. The scaling curves are fitted on the validation points, which measure the average pass rate every 100 training steps, with 16 generations per prompt on the $1,000$ held-out prompts.

## 3 AN EMPIRICAL STUDY OF RL SCALING

In this section, we conduct RL experiments using an 8B dense model on verifiable math problems. Using the setup described in Section 2, we study several design axes in terms of their predictable compute-scaling behavior, namely *asymptotic performance* ($A$) and *compute efficiency* ($B$), as shown in Figure 3.

We structure our experiments in three stages – we first ablate design choices on top of the baseline at $3.5k$ to $4k$ GPU-hours since some experimental choices destabilize beyond this scale (Appendix A.16). Whenever a design change proved stable, we trained it for longer. Then, we combine the best choices into **SCALERL** and run leave-one-out (LOO) experiments for $16k$ GPU-hours in Section 4. Here, we assess predictability by fitting on the first $8k$ GPU-hours and extrapolating the remainder of the run. Finally, to demonstrate predictable scaling with **SCALERL**, we also consider training setups with larger batch sizes, mixture-of-experts model, multiple tasks (math and code), and longer sequence lengths in Section 5.

### 3.1 ASYNCHRONOUS RL SETUP

We first investigate the choice of asynchronous off-policy RL setup (Noukhovitch et al., 2024), as it governs training stability and efficiency, generally independent of other design choices. Specifically, we consider two approaches for off-policy learning: PPO-off-policy-$k$ and PipelineRL-$k$.

**PPO-off-policy-**$k$ is the default approach for asynchronous RL and has been used previously by Qwen3 (Yang et al., 2025) and ProRL (Liu et al., 2025a). In this setup, the old policy $\pi_{gen}^{\theta_{\text{old}}}$ generates reasoning traces for a batch of $B$ prompts. Each gradient update processes a mini-batch of $\hat{B}$ prompts, resulting in $k = B/\hat{B}$ gradient updates per batch. In our experiments, we fix $\hat{B} = 48$ prompts (with 16 generations each), and vary $k \in \{1, 8\}$ by setting $B = k \times 48$.

**PipelineRL-**$k$ is a recent approach from Piche et al. (2025) and used by Magistral (Rastogi et al., 2025). In this regimen, generators continuously produce reasoning traces in a streaming fashion. Whenever trainers finish a policy update, the new parameters are immediately pushed to the generators, which continue generating with the updated weights but a stale KV cache from the old policy. Once a full batch of traces is generated, it is passed to the trainers for the next update. In our setup we introduce a parameter $k$: the trainers wait if they get $k$ steps ahead of the generators.

We compare these approaches in Figure 4a. PipelineRL and PPO-off-policy achieve similar asymptotic performance $A$, but PipelineRL substantially improves the compute efficiency $B$; thus reaching the ceiling $A$ faster. This is because PipelineRL reduces the amount of idle time in the training process. This choice yields reliable gains with fewer tokens, making larger sweeps at a lower compute budget possible. We also vary the maximum off-policyness for PipelineRL and find $k = 8$ to be optimal, discussed in Appendix A.12.

## 3.2 Algorithmic Choices

Building on the results above, we adopt PipelineRL-8 as our updated baseline. We then study six additional algorithmic axes: (a) loss aggregation, (b) advantage normalization, (c) precision fixes, (d) data curriculum, (e) batch definition, and (f) loss type. In Section 4, we combine the best options into a unified recipe, termed **SCALERL**(**Scale**-able **RL**), and conduct leave-one-out experiments on a larger scale of $16,000$ GPU-Hours.

**Loss type** We compare the asymmetric DAPO loss (Eq. 8) with two recently proposed alternatives: GSPO (Zheng et al., 2025a) and CISPO (MiniMax et al., 2025; Yao et al., 2025). GSPO applies importance sampling at the sequence level as opposed to GRPO's token-level formulation. Specifically, GSPO alters the token-level IS ratio (Eq. 2) to sequence-level ratios: $\rho_i(\theta) = \pi_{train}(y_i|x,\theta)/\pi_{gen}(y_i|x,\theta_{\text{old}})$. CISPO simply combines truncated IS with vanilla policy gradient (Ionides, 2008), where $\mathrm{sg}$ is the stop-gradient function:

$$\mathcal{J}_{\text{CISPO}}(\theta) = \mathbb{E}_{\substack{x \sim D, \\ \{y_i\}_{i=1}^{G} \sim \pi_{gen}(\cdot|x,\theta_{\text{old}})}} \left[ \frac{1}{T} \sum_{i=1}^{G} \sum_{t=1}^{|y_i|} \mathrm{sg}(\min(\rho_{i,t}, \epsilon_{\max})) \hat{A}_i \log\left(\pi_{train}(y_{i,t}|x, y_{i<t}, \theta)\right) \right] \quad (4)$$

Figure 4b shows that both GSPO and CISPO substantially outperform DAPO, improving the asymptotic pass-rate $A$ by a large margin. CISPO exhibits a prolonged near-linear reward increase, and is marginally better than GSPO later in training, so we opt for CISPO as our best loss type. Further discussion on off-policy loss types, and their hyperparameter robustness is detailed in Section 4 and Appendix A.18.

**FP32 Precision for LLM logits** The generators and trainers rely on different kernels for inference and training, leading to small numerical mismatches in their token probabilities (He & Lab, 2025). RL training is highly sensitive to such discrepancies, since they directly affect the IS ratio in the surrogate objective. MiniMax et al. (2025) identified that these mismatches are especially pronounced at the language model head, and mitigate this by FP32 computations at the head for both the generator and trainer. As shown in Figure 4c, the precision fix dramatically improves the asymptotic performance $A$ from 0.52 to 0.61, and inclyde it in our **SCALERL** recipe.

**Loss Aggregation** We evaluate three strategies for aggregating the RL loss: (a) *Sample average* where each rollout contributes equally (as in GRPO, Appendix A.2). (b) *Prompt average* where each prompt contributes equally (as in DAPO, Appendix A.2). (c) *Token average* where all token losses in the batch are averaged directly, without intermediate grouping. The comparison results are shown in Appendix A.9 (Figure 10a). We find prompt-average achieves the highest asymptotic performance and therefore use this choice for **SCALERL**.

**Advantage Normalization** We compare three variants of advantage normalization: (a) *Prompt level* where advantages are normalized by the standard deviation of rewards from the rollouts of the same

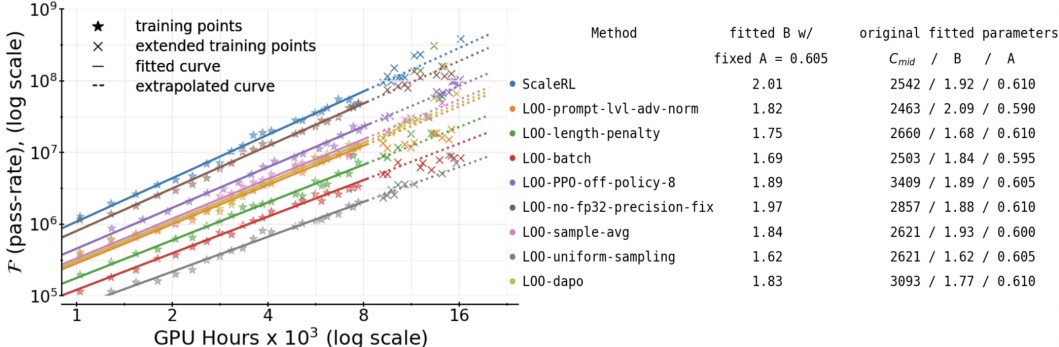

Figure 5: **Leave-One-Out (LOO) Experiments**: Starting from **SCALERL**, we revert one design choice at a time to its baseline counterpart and re-train. Most LOO variants reach a similar asymptotic reward, with **SCALERL** outperforming slightly overall. The main difference in these methods lies in efficiency. To highlight this, we re-arrange Equation (1) into $\mathcal{F}(R_c) = C^B$, where $\mathcal{F}(R_c) = C_{\text{mid}}^B / \left( \frac{A-R_0}{R_c-R_0} - 1 \right)$, and plot $\log \mathcal{F}(R_c)$ vs. $\log C$. This makes slope $B$ directly visible, showing that **SCALERL** has the highest compute efficiency.

prompt (as in GRPO, Appendix A.2). (b) *Batch level* where advantages are normalized by the standard deviation across all generations in the batch, as used by Hu et al. (2025a); Rastogi et al. (2025). (c) *No normalization* where advantages are computed as raw rewards centered by the mean reward of the prompt's generations, without variance scaling (as proposed in Dr. GRPO (Liu et al., 2025b)). A comparison plot is shown in Appendix A.9 (Figure 10b), and all three methods are oberseved to yield similar performance. We therefore adopt batch-level normalization as it is theoretically sound and marginally better.

**Zero-Variance Filtering** Within each batch, some prompts yield identical rewards across all their generations. These "zero-variance" prompts have zero advantage and therefore contribute zero policy gradient. The default baseline includes such prompts in loss computation, but it is unclear whether they should be included in the effective batch. To test this, we compare the default setting against an *effective batch* approach, where only prompts with non-zero variance are included in the loss calculation, as done by Seed et al. (2025). Note that zero-variance filtering differs from dynamic sampling in DAPO (Yu et al., 2025). The former merely drop the prompts, while latter resamples more prompts until the batch is full. We show in Appendix A.9 (Figure 11a) that using the effective batch performs better asymptotically; and we adopt it in our **SCALERL** recipe.

**Adaptive Prompt Filtering** A number of data curriculum strategies have been proposed for RL training to improve sample efficiency (An et al., 2025; Zhang et al., 2025b; Zheng et al., 2025b). Here we evaluate a simple variant, introduced by An et al. (2025), with the key observation that once a prompt becomes too easy for a policy, it typically remains easy. Since such prompts consume some compute but no longer contribute useful gradient signal, it is better to exclude them from future training. We implement this by maintaining a history of pass rates and permanently removing any prompt with pass rate $\geq 0.9$ from subsequent epochs–we call this **No-Positive-Resampling**. In Appendix A.9 (Figure 11b) we compare this curriculum against the default setting where all prompts are resampled uniformly throughout training. We see that the curriculum improves scalability and the asymptotic reward $A$.

# 4 SCALERL: SCALING RL COMPUTE EFFECTIVELY & PREDICTABLY

From the design axes studied above, we consolidate the best-performing settings into a single recipe, which we term **SCALERL** (**Scale**-able **RL**). **SCALERL** is an asynchronous RL recipe that uses **PipelineRL with** 8 **steps off-policyness**, **interruption-based length control** for truncation, **FP32 computation for logits**, and optimizes the $\mathcal{J}_{\textbf{SCALERL}}(\theta)$ **loss**. This loss combines prompt-level loss aggregation, batch-level advantage normalization, **truncated importance-sampling** REINFORCE

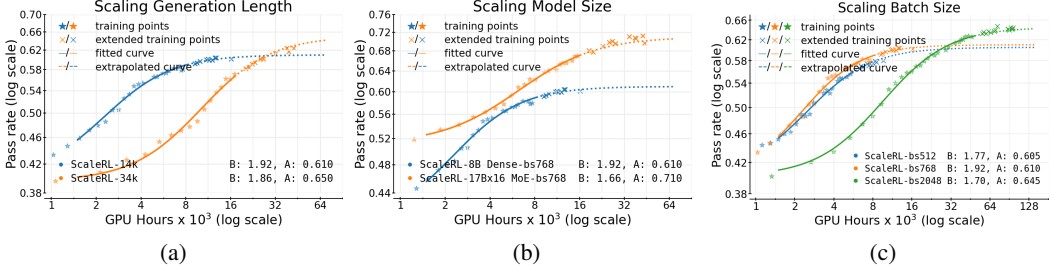

Figure 6: **Scaling RL across several axes**. We scale up the ScaleRL recipe across three axes: (a) *sequence length:* Larger equence is slower but reaches a higher asymptotic reward, (b) *model size:* using the larger 17Bx16 Llama-4 model reaches a much higher terminal reward and shows better generalization, and (c) *batch size:* larger batch is slower in training but settles at a higher asymptote.

loss (CISPO) , zero-variance filtering, and no-positive resampling:

$$
\mathcal{J}_{\textbf{SCALERL}}(\theta) = \mathop{\mathbb{E}}_{\substack{x \sim D, \\ \{y_i\}_{i=1}^{G} \sim \pi_{gen}^{\theta_{old}}(\cdot|x)}} \left[ \frac{1}{\sum_{g=1}^{G} |y_g|} \sum_{i=1}^{G} \sum_{t=1}^{|y_i|} \text{sg}(\min(\rho_{i,t}, \epsilon)) \hat{A}_i^{\text{norm}} \log \pi_{train}^{\theta}(y_{i,t}) \right],
$$

$$
\rho_{i,t} = \frac{\pi_{train}^{\theta}(y_{i,t})}{\pi_{gen}^{\theta_{old}}(y_{i,t})}, \ \hat{A}_i^{\text{norm}} = \hat{A}_i / \hat{A}_{\text{std}}, \ 0 < \text{mean}(\{r_j\}_{j=1}^{G}) < 1, \ \text{pass\_rate}(x) < 0.9,
$$

where $\text{sg}$ is the stop-gradient function, $\hat{A}_{\text{std}}$ is the standard deviation of all advantages $\hat{A}_i$ in a batch and pass_rate$(x)$ is the historical pass rate of prompt $x$. For forced interruptions, we use phrase: "Okay, time is up. Let me stop thinking and formulate an answer now. $</\text{think}>$".

**Leave-One-Out (LOO) Ablations** To validate that these choices remain optimal when combined, we conduct *leave-one-out* (LOO) experiments: starting from **SCALERL**, we revert one axis at a time to its baseline counterpart from Section 2. This ensures that each decision contributes positively even in the presence of all others. Figure 5 reports these experiments, each scaled to 16k GPU hours.

Across all axes, **SCALERL** consistently remains the most effective configuration, slightly outperforming LOO variants either in asymptotic reward or in compute efficiency (refer to the last column in the Figure 5 table). Since most LOO variants reach similar asymptotic pass rates, we transform the sigmoidal fit to a power-law fit, to highlight efficiency differences via the slope $B$ (details in Figure 5). Concretely, we average the asymptotic reward $A$ across all runs, re-fit the curves with this fixed $A$, and then compare slopes (measuring efficiency) in Figure 5.

In all our LOO experiments as well as independent **SCALERL** runs, we fit the sigmoidal curve up to 8k GPU-hours and extrapolate to 16k GPU-hours, observing that the predicted curves align closely with both training and extended points. This demonstrates the stability and predictability of **SCALERL** and other stable, scalable recipes under large-scale RL training.

## 5 PREDICTABLE SCALING RETURNS ACROSS RL COMPUTE AXES

Given a fixed or growing compute budget, which scaling knob –context length, batch size, generations per prompt, and model size – buys the most reliable performance gain, and how early can we predict that return? We answer this by (i) fitting the saturating power-law in eq. (1) early in training for each setting (precisely, half the target budget), (ii) extrapolating to the target budget, and (iii) extending training to verify the forecast. Across all axes below we observe clean, predictive fits whose extrapolated curves align with the extended trajectories, mirroring the behavior seen in our 100,000 GPU-hour run (Figure 1), and the cross-recipe comparison in Figure 2.

**Model scale (MoE)** Does ScaleRL remain predictive and stable on larger models? Training the 17B×16 Llama-4 Scout MoE with ScaleRL exhibits the same predictable scaling behavior as the 8B model, with low truncation rates and no instability pathologies (Appendix A.16, A.18). Figure 1 shows the training curve. The extended points align with the fitted curve, supporting the model-scale

invariance of our recipe. Moreover, the larger 17B×16 MoE exhibits much higher asymptotic RL performance than the 8B dense model, outperforming the 8B's performance using only $1/6$ of its RL training compute.

**Generation length (context budget)** Increasing the generation length from 14k to 32k tokens slows early progress (lower $B$ and higher $C_{mid}$) but consistently lifts the fitted asymptote (A), yielding higher final performance once sufficient compute is provided (Figure 6). This validates long-context RL as a ceiling-raising knob rather than a mere efficiency trade-off. Extrapolations made from the fit correctly forecast the higher 32k-token trajectory when training is extended.

**Global batch size (prompts)** Smaller-batch runs show early stagnation on downstream benchmarks even as in-distribution validation performance continues to improve. Larger batches reliably improve the asymptote and avoid the downstream stagnation we observe in smaller-batch runs. Figure 6c shows the same qualitative pattern at mid-scale: small batches may appear better early but are overtaken as compute grows. In our largest run in Figure 1, moving to batch size of 2k both stabilized training and yielded a fit that extrapolated from 50k GPU hours to the final 100k point.

**Generations per prompt (fixed total batch)** For a fixed total batch, is it better to allocate more prompts or more generations per prompt? Sweeping generations per prompt 8,16,24,32 and adjusting prompts to keep total batch fixed leaves fitted scaling curves essentially unchanged (Appendix A.14), suggesting that, at moderate batch, this allocation is a second-order choice for both A and B. Clearer differences may emerge at much larger batches, which we leave as future work.

## 6 RELATED WORK

We detail two most relevant works to our study in this section. ProRL (Liu et al., 2025a) demonstrates that prolonged RL fine-tuning on LLMs ($\sim$ 2000 optimization steps, 64 batch size) for 16K GPU-hours using a mix of reasoning tasks uncovers novel solution strategies beyond a model's base capabilities. This longer training regimen delivered significant gains on a 1.5B model, rivaling the performance of larger models on some benchmarks. ProRL's contributions lie in specific heuristics for stability (KL-regularization, policy resetting, entropy controls, etc.) to achieve high performance out of a 1.5B model.

Liu et al. (2025c) offer a complementary perspective and ablates various design choices under consistent conditions on Qwen-3 4B/8B (Yang et al., 2025), and presents a minimalist combination, LitePPO, that outperforms more complex methods like GRPO (Shao et al., 2024) and DAPO (Yu et al., 2025) on smaller scale models and compute. This yields valuable algorithmic insights, but the focus is on comparative empirical findings, rather than on scaling behaviour. While we focus on defining universal scaling trajectories for reasoning tasks, Vattikonda et al. (2026) has employed extensive hyperparameter sampling and bootstrapping to provide a 'statistical diagnosis' of RL training recipes in domain-specific contexts like web-agent post-training.

None of these work study "scaling" properties of these methods. In fact, the main comparisons are done on downstream evaluations, which may not be not the right metric to study predictable scaling. Rather, as done in pre-training and in our work here, we study performance on in-distribution held out eval set. In contrast to the mentioned related works, our work develops and validates a compute-performance framework with predictive fits, while operating at a much larger compute budget (e.g, $6x$ larger than ProRL) and model scale compared to the above studies. Additionally, our findings yield a near state-of-the-art RL recipe that can scale *predictably* to over 100,000 GPU-hours without any stability issues. The rest of the related work is deferred to Appendix A.1.

## 7 DISCUSSION & CONCLUSION

In this work, we study the scaling properties of different techniques used in RL for LLMs in pursuit of a predictable scalable recipe. With this mission, we derive a method for fitting predictive scaling fits for accuracy on the validation set that allows us to quantify the asymptotic performance and compute efficiency of an RL method. Using this methodology, our primary contribution is to conduct a careful series of ablations of several algorithmic options that go into the RL recipe. For each ablation, we choose the option with higher asymptotic performance when possible and improved

efficiency otherwise. Combining these choices yields the **SCALERL** recipe which scales better than all existing recipes in our experiments. A few observations are in order:

- **Compute scaling extrapolation**. An important insight of our scaling methodology is that we can use smaller-scale ablations in a systematic way to predict performance at larger scales. This allows us to create our final scalable recipe.

- **Most important decisions**. The off-policy algorithm, loss function, and model precision are the most important decisions from our ablations. Each of the other decisions does not have a large individual effect, but as we see from the leave-one-out experiments, they still do have some cumulative impact (in terms of efficiency) when all combined.

- **Asymptotic performance vs. efficiency**. For many of our ablations, we found the better option to improve both efficiency and asymptotic performance, but this is not always the case (e.g. for FP32, Figure 4c). When doing the "forward" ablations starting from the baseline method, we opt for asymptotic performance first and foremost. Interestingly, when doing the "backward" leave-one-out ablations from the **SCALERL** recipe, we find very little impact on asymptotic performance from each decision, but each component of the algorithm seems to help efficiency. This shows that the cumulative effect of the changes is quite robust.

- **Generalization**. While we report transfer to downstream evaluations, our primary focus is on studying predictive scaling, which is characterized through in-distribution performance curves on a held-out dataset from training prompts (Li et al., 2025b; Muennighoff et al., 2025). This still leaves the question of how well the LLM would generalize from the training distribution to held out test sets. While a full characterization of generalization is beyond the scope of our work, we do observe correlation between in-distribution validation and downstream generalization performance. However, there are some algorithmic choices that seem to help generalization more, that we want to note here including: larger batch size (Section A.15), reducing truncations (Section A.16), longer generation lengths (Figure 17b), and larger model scale (Section 5, Figure 1).

- **Multi-task RL**. While our experiments focus mainly on the math domain, we also evaluate **SCALERL** under multi-task RL training. As shown in Figure 16, joint training on math and code yields clean, parallel power-law trends for each domain, with extended runs remaining aligned with the extrapolated curves. While our preliminary results are promising, it would be interesting to thoroughly study predictability of compute scaling for multi-task RL with different training data mixtures.

**Future work** A natural next step is to derive predictive "scaling laws" for RL across pre-training compute, model size, and RL training data. Future studies can also include other axes of RL compute scaling, such as incorporating structured or dense rewards (Setlur et al., 2024) and more compute-intensive generative verifiers (Zhang et al., 2025a), to find optimal compute allocation for RL training. Finally, the framework introduced here can be applied to study the scaling behavior of other post-training regimes, including multi-turn RL, agentic interaction, and long-form reasoning.

There are of course many design choices in RL, so we don't think that our **SCALERL** recipe is the end of the story. We hope that our focus on scalable RL and methodology for predicting scalability can inspire future work to push the frontier of RL for LLMs even further. To enable future studies to fit compute-performance RL scaling curves, we release a minimal code repository at www.devvrit.com/scalerl_curve_fitting.

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

## A APPENDIX

### A.1 EXTENDED RELATED WORK

A wave of recent work has applied Reinforcement Learning (RL) to improve the reasoning abilities of large language models (LLMs); often achieving state-of-the-art results on challenging tasks (OpenAI, 2024; Guo et al., 2025; Seed et al., 2025; Carbonneaux et al., 2025). OpenAI's o1 series of models established that large-scale RL can substantially enhance long-horizon reasoning, but did not release any details on how these models were trained. Deepseek R1 (and R1-Zero) (Guo et al., 2025) provided the first comprehensive study on training high-performing and long Chain-of-Thought (CoT) models primarily via RL, documenting emergent behaviours under extended RL without any reliance on reward models (Lightman et al., 2023) or Monte Carlo Tree Search (MCTS) (Xie et al., 2024).

The earliest widely referenced RLVR (verifiable-reward) algorithm underlying this wave of reasoning development is Group Relative Policy Optimization (GRPO), introduced in Shao et al. (2024). GRPO is a critic-free, group-relative policy gradient with PPO-style clipping that replaces a learned value baseline with group baselines to reduce computational cost and stabilize credit assignment for long CoTs. While GRPO catalyzed rapid progress, subsequent work document its limitations (token-level clipping, model collapse risks) and motivate different group- or sequence- level variants (Yu et al., 2025; Yue et al., 2025; Hu et al., 2025b; Zheng et al., 2025a).

Yu et al. (2025) propose the Decoupled clip and Dynamic Sampling Policy Optimization (DAPO), where they decouple $\epsilon_{\texttt{low}}$ and $\epsilon_{\texttt{high}}$ clipping in the GRPO objective and do *Clip-Higher* for $\epsilon_{\texttt{high}}$ to avoid entropy collapse. Furthermore, they do dynamic sampling of prompts in a given batch to avoid samples with zero variance (or advantage) which contribute zero policy gradients. Finally, they employ token-level loss aggregation unlike GRPO, which uses sample-level loss averaging. With these modifications, they are able to surpass the vanilla GRPO baseline while avoiding entropy collapse in the RL training. In parallel, Yue et al. (2025) develop VAPO; a value-augmented PPO tailored for long CoTs with strong stability and outperforming value-free baselines like GRPO and DAPO. They combine value pre-training and decoupled Generalized Advantage Estimation (GAE) from VC-PPO (Yuan et al., 2025), loss objective modifications from DAPO, and propose length-adaptive GAE to come up with an open recipe, VAPO, that has been used to train large MoE models in Seed et al. (2025). Similarly, other technical report like Magistral (Rastogi et al., 2025), Kimi-k1.5 (Kimi Team et al., 2025b), Minimax-01 (Li et al., 2025a) detail various details on their RL training recipes, but don't share extensive experiments on why their design choices are better than the baselines.

### A.2 RL FOR LLMS: GRPO AND DAPO

**Group Relative Policy Optimization (GRPO)** GRPO (Shao et al., 2024) adapts PPO Schulman et al. (2017) for LLM fine-tuning with verifiable rewards. For a given prompt $x$, the old policy $\pi_{\text{gen}}(\theta_{\text{old}})$ generates $G$ candidate completions $\{y_i\}_{i=1}^G$, each assigned a scalar reward $r_i$. To emphasize relative quality within the group, rewards are normalized as

$$\hat{A}_i = \frac{r_i - \text{mean}(\{r_j\}_{j=1}^G)}{\text{std}(\{r_j\}_{j=1}^G) + \epsilon}. \tag{5}$$

Each completion $y_i$ of length $|y_i|$ contributes at the token level through ratios

$$\rho_{i,t}(\theta) = \frac{\pi_{train}(y_{i,t} \mid x, y_{i,<t}, \theta)}{\pi_{gen}(y_{i,t} \mid x, y_{i,<t}, \theta_{\text{old}})} \tag{6}$$

The GRPO objective averages across both completions and tokens:

$$\mathcal{J}_{\text{GRPO}}(\theta) = \mathbb{E}_{\substack{x \sim D, \\ \{y_i\}_{i=1}^G \sim \pi_{\text{gen}}(\cdot \mid x, \theta_{\text{old}})}} \left[ \frac{1}{G} \sum_{i=1}^G \frac{1}{|y_i|} \sum_{t=1}^{|y_i|} \min\left( \rho_{i,t}(\theta) \hat{A}_i, \ \text{clip}(\rho_{i,t}(\theta), 1 \pm \epsilon) \hat{A}_i \right) \right] \tag{7}$$

Thus GRPO preserves token-level policy ratios as in PPO, while using sequence-level, group-normalized advantages to stabilize learning under sparse rewards.

**Decoupled Clip and Dynamic Sampling Policy Optimization (DAPO)** DAPO (Yu et al., 2025) extends GRPO with two key modifications. First, it replaces symmetric clipping with *asymmetric clipping*, using distinct thresholds for upward and downward deviations: $\text{clip}_{\text{asym}}(\rho, a) = \text{clip}(\rho, 1 - \epsilon^-, 1 + \epsilon^+)$, where $\epsilon^-$ and $\epsilon^+$ are hyper-parameters.

Second, DAPO changes the aggregation scheme to operate at the *prompt level*. For a given prompt $x \sim D$, the old policy produces $G$ completions $\{y_i\}_{i=1}^G$ with advantages $\{\hat{A}_i\}$ (Equation (5)). Let $T = \sum_{i=1}^G |y_i|$ denote the total number of tokens across all completions. With token-level ratios as in Equation (2). The DAPO surrogate objective is

$$\mathcal{J}_{\text{DAPO}}(\theta) = \mathbb{E}_{\substack{x \sim D, \\ \{y_i\}_{i=1}^G \sim \pi_{\text{gen}}(\cdot | x, \theta_{\text{old}})}} \left[ \frac{1}{T} \sum_{i=1}^G \sum_{t=1}^{|y_i|} \min\left( \rho_{i,t}(\theta)\hat{A}_i, \ \text{clip}_{\text{asym}}(\rho_{i,t}(\theta))\hat{A}_i \right) \right]. \quad (8)$$

This prompt-level normalization ensures that each token contributes equally to the prompt's loss, regardless of the number or length of its sampled completions. DAPO also introduces dynamically dropping 0-variance prompts from the batch during training and filling the batch with more prompts until the batch is full. We skip that change here since its effect is similar to having a larger batch size.

### A.3 TRAINING SETUP

**Datasets** For small-scale SFT, we use a curated data mix of reasoning traces. We filter this dataset by removing trivial prompts, discarding solution traces exceeding $12k$ tokens, and decontaminating with AIME 2024/2025 (AoPS, 2025) and MATH-500 (Hendrycks et al., 2021) benchmarks. For the RL stage, we use the Polaris-53K dataset (An et al., 2025) for most of our runs; additionally using the Deepcoder dataset (Luo et al., 2025) for runs with both math and code.

**Supervised Fine-tuning** We run SFT using a batch size of 2M tokens, max sequence length of 12288, and a learning rate of $3 \times 10^{-5}$ using the AdamW optimizer (Loshchilov & Hutter, 2019) on 32 H100 GPU nodes for approximately 4 epochs and 32B tokens in total.

**Reinforcement Learning** We allocate 14k generation budget during RL training, where 12k tokens are allocated to the intermediate reasoning ("thinking"), followed by 2k tokens for the final solution and answer. We sample $48$ prompts in each batch, each with $16$ generations per prompt. Thus, we get the total batch size as 768 completions per gradient update step. The rewards are given as $\pm 1$ to correct and incorrect traces respectively. We use a constant learning rate of $5 \times 10^{-7}$, AdamW optimizer (Loshchilov & Hutter, 2019) with $\epsilon = 10^{-15}$, weight decay of 0.01 (default in AdamW), and a linear warmup of 100 steps. The lower $\epsilon$ is to avoid gradient clipping (epsilon underflow) (Wortsman et al., 2023).

We use automated checkers like Sympy (Meurer et al., 2017) or Math-Verify[1] for assessing the correctness of the final answer for math problems after stripping out the thinking trace (`<think>`···`</think>`). We use a custom code execution environment for coding problems involving unit tests and desired outputs.

We used 80 Nvidia GB200 GPU for a single run, with a compute budget ranging from 3.5-4K GPU hours for establishing different design choices in Section 3.2, 16K for the leave-one-out experiments (Section 4), and finally 30k-100K GPU hours for our larger scale runs (Section 5). We adopt a generator–trainer split between GPUs. For 80 GPU experiments, we set $64$ of those as *generators*, responsible for the generation of reasoning trace using the optimized inference codebase. The remaining 16 GPUs act as *trainers*, which receive generated trajectories, perform policy updates, and periodically broadcast updated parameters back to the generators.

**Training Regimen** All experiments are conducted on the RL for reasoning domain, where the model produces a thinking trace enclosed with special tokens (`<think>` ... `</think>`) and a final solution. Unless noted, training uses a sequence length of $16,384$ tokens: $12,288$ for thinking, $2,048$ for the solution, and an additional $2,048$ for the input prompt. We adopt the $12,288$ thinking

---

[1]https://github.com/huggingface/Math-Verify

budget for faster iteration, and show in Section 5 that **SCALERL** extrapolations remain predictive when training with larger thinking budgets (32, 768). For math RL experiments, we use the Polaris-53K dataset (An et al., 2025) with a batch size of 768 (48 prompts with 16 generations each). In our setup, scaling RL compute corresponds to running multiple epochs over the training prompts.

## A.4 What curve to fit?

Pre-training curves are usually fit using power-law equation (Li et al., 2025b; Kaplan et al., 2020; Muennighoff et al., 2025), which in our case would model performance as $R_C = A - D/C^B, C \geq C_0$, where $D$ is a constant, and $C_0$ marks the compute threshold beyond which the law holds. Intuitively, this implies that each multiplicative increase in compute yields a constant proportional gain in performance. For RL post-training, however, we find a sigmoidal fit (eq. (1)) more appropriate for several reasons. First, for bounded metrics such as accuracy or reward, sigmoidal curves provide better predictive fits (Ruan et al., 2024; Srivastava et al., 2022); we observe the same, with accurate extrapolation to higher compute (Figure 1). Second, power laws are unbounded at low compute and are typically fit only beyond a threshold $C_0$. In RL, where total training spans far fewer steps (e.g., only $\sim$75 evaluation points to fit only in Figure 1), discarding early points a lot further reduces the already limited data available for fitting. Third, empirically, sigmoidal fits are substantially more robust and stable than power-law fits. Concretely, consider the 100k GPU-hour run on the 8B dense model shown in Figure 1. When we fit a power-law curve between 1.5k–50k GPU hours, it predicts an asymptotic performance of $A = 1.0$, which is clearly incorrect - the actual curve saturates near 0.65. In contrast, the sigmoidal fit yields an accurate prediction of $A = 0.645$. Moreover, the power-law fit is highly sensitive to the chosen fitting regime: fitting over (5k, 50k) GPU hours instead gives $A = 0.74$, while the sigmoidal fit remains robust and still predicts $A = 0.645$. Power-law models only recover the correct asymptote when fitted exclusively in the high-compute regime (e.g., 30k–60k GPU hours). However, our goal is to predict large-scale performance from lower-compute regimes, where such long runs are unavailable.

Given these considerations, we use the sigmoidal form throughout our analysis. Intuitively, a sigmoidal curve captures saturating returns - grows slowly in the low-compute regime, accelerates sharply through a mid-range of efficient scaling, and then saturates at high compute as it approaches a finite performance ceiling.

One thing to note is that at high compute regime, sigmoidal curve behaves same as power-law. Concretely, we can have the following approximation of sigmoidal curve:

$$R_C = R_0 + \frac{A - R_0}{1 + (C_{mid}/C)^B} \qquad \text{(sigmoidal curve from eq. (1))}$$

$$\implies R_C \approx R_0 + (A - R_0)\left(1 - \frac{C_{mid}^B}{C^B}\right) \qquad \text{(For } C >> C_{mid}, \text{ high compute regime)}$$

$$= A - \frac{(A - R_0)C_{mid}^B}{C^B}$$

$$= A - \frac{D}{C^B}$$

where above $D = (A - R_0)C_{mid}^B$. And this is the same form of power-law mentioned at the start of this section.

## A.5 Fitting scaling curves

We fit the sigmoid-law equation in Equation (1) to the mean reward on our held-out validation set. This set consists of $1,000$ prompts held out from the Polaris-53k (An et al., 2025) math dataset, with 16 generations sampled every evaluation step performed at 100 steps intervals.

Directly fitting all three parameters $\{A, B, C_{mid}\}$ is challenging. Instead, we perform a grid search over $A \in \{0.450, 0.455, 0.460, \dots, 0.800\}$ and $C_{mid} \in [100, 40000]$ (searching over 100 linearly separated values), and for each candidate $A, C_{mid}$ fit $B$. The best fit (measured by sum of squared residuals) across this grid is selected as the final curve. We use SciPy's `curve_fit` with default initialization; varying the initialization strategies produced identical results. To enable future studies

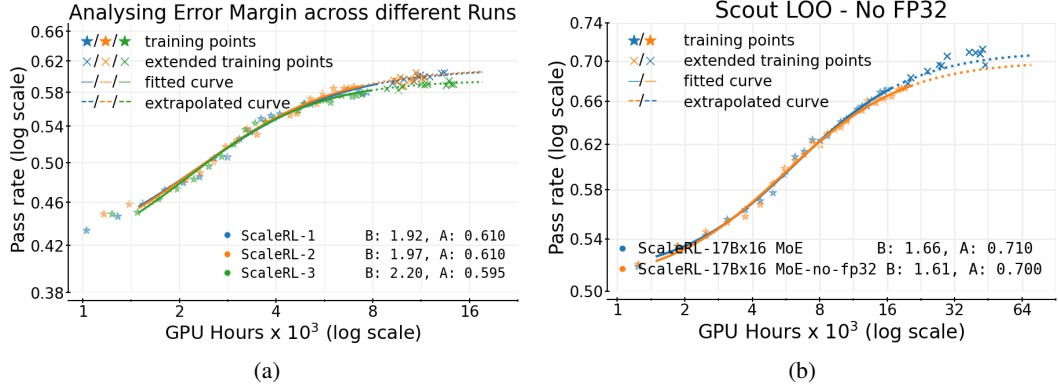

(a)                                                (b)

Figure 7: (a) **Variance in scaling fits**. We train 3 independent runs of **SCALERL** to measure variance. We observe a $\pm 0.02$ error margin for asymptotic performance $A$. (b) **FP32 LOO on Scout:** Comparing **SCALERL** on Scout with and without FP32 precision fix at the LM Head. **SCALERL** performs better with the FP32 fix.

to fit compute–performance RL scaling curves, we promise to release a code repository after the acceptance of the paper.

To estimate the error margin of our fits, we trained three independent **SCALERL** runs with a batch size of $768$ and generation length of $14k$ (as used in Section 4), shown in Figure 7a. We found that the fit values of $A$ varied by at most $\pm 0.015$, suggesting $0.02$ as a reasonable error margin on the estimates of asymptotic performance. Estimating the error margin for the fitted value $B$ is difficult, as different algorithms with different $A$ values can have different error margins for $B$. However, for the purpose of comparing algorithms, we can safely deduce that if two methods achieve similar $A$ values (within $0.02$), the one with higher $B$ when a refit is done with the average of $A$ values is at least as good in terms of efficient scalability.

### A.6 COMPARING ALGORITHMS

Consistent with observations in large-scale pre-training, where the loss exhibits a sharp initial drop before settling into a predictable power-law decay (Li et al., 2025b), we observe a similar two-phase behavior in RL. The mean reward increases rapidly, almost linearly, during the $\sim$first epoch ($\sim$ 1k steps, or $\sim$1.5k GPU Hours for most runs), after which the curve follows sigmoidal-law behavior (see Figure 12 to see the "sigmoid" like curve). Our sigmoidal-law fits are applied to this latter portion of the training curve.

Unlike pre-training, our main goal is not to predict the performance of a fixed recipe, but to identify which algorithms and design choices scale reliably, and to design algorithm that exhibits predictive nature. Achieving highly robust fits typically requires very large runs with hundreds or thousands of evaluation points, which is impractical in our setting for two reasons. First, running all ablations at such scale would be computationally prohibitive. Second, many RL algorithms we compare are themselves not scalable to such extreme budgets: they often saturate much earlier or even degrade with more compute due to instability. For example, our baseline method (Section 3.2) destabilizes beyond $\sim$ 3500 GPU-hours, since overlong generation truncations exceed $10\%$ of generations - reducing the effective batch size. More discussion on this is in Section A.16.

As we ablate across different axes in Section 3.2, we discover design choices that improve stability at higher compute. Some ablated variants can scale further, e.g., $\sim$ 5k GPU hours for $\epsilon = 0.26$ in DAPO, $\sim$ 6k GPU hours with the FP32 precision fix, and $\sim$ 7k GPU hours for CISPO. Once we combine the best design choices, we obtain a stable and scalable recipe, which allows us to run leave-one-out (LOO) experiments for $\sim$ 1600 GPU hours per run.

## A.7 ROBUSTNESS OF FITS

One may wonder how robust our fitted curves are. We address a few relevant points below:

- For stable and scalable experiments, including all runs from Section 4 onward, changing the fitting regime (e.g., including or excluding the initial 1.5k GPU-hour range) yields similar predictable results. For instance, in the 100k GPU-hour run on the 8B dense model, fitting over $(1.5k, 50k)$ gives $B = 1.70$, $A = 0.645$, while $(0, 100k)$ gives $B = 1.56$, $A = 0.655$, $(0, 50k)$ gets $B = 1.7$, $A = 0.645$, and $(5k, 50k)$ gives $B = 1.67$, $A = 0.645$. Across these regimes, parameter values remain within the expected error margin (Section 7).

- We nonetheless skip the low-compute regime because early training phases, especially in less stable setups from Section 3.2, often plateau prematurely or deviate from the sigmoidal trend due to transient instabilities (see Appendix A.6, A.16). Excluding this region allows the fit to focus on the mid-to-high compute range where saturation behavior is clearer and more consistent.

- The 1.5k GPU-hour threshold is a heuristic chosen empirically: it approximately corresponds to one epoch for most experiments in Section 3.2. Larger cutoffs reduced the number of fitting points, while smaller ones often introduced noise. We found 1.5k GPU hours to provide the best balance between fit stability and sample coverage, consistent with practices of skipping low-FLOPs regime in pre-training scaling analyses and fitting (Li et al., 2025b).

## A.8 INTERPRETING SIGMOIDAL CURVES

Figure 3 presented an example fit illustrating the influence of parameters $A$, $B$, and $C_{\text{mid}}$. Here, we extend this with additional illustrations: Figure 8a, Figure 8b, and Figure 9a vary $B$, $C_{\text{mid}}$, and $A$ respectively, while keeping the other parameters fixed. We observe that $B$ and $C_{\text{mid}}$ primarily affect the *efficiency* of scaling, whereas $A$ determines the asymptotic performance achievable at large compute. In Figure 9b we see a case of two runs where one is much more efficient, hence shows initial promising gains, but converges to a lower asymptote, while the other progresses more slowly yet ultimately surpasses it due to a higher $A$. In practice, scaling strategies should prioritize design choices that raise the asymptotic ceiling $A$, and only then optimize for efficiency parameters such as $B$ or $C_{\text{mid}}$.

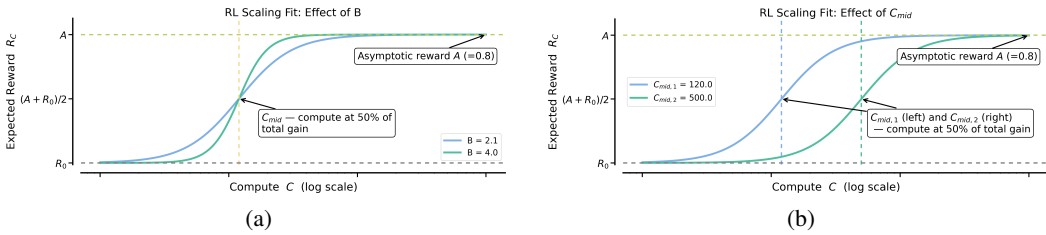

Figure 8: Keeping all parameters same and only changing (a) $B$, (b) $C_{mid}$. Both these parameters modulate the efficiency of the training run.

## A.9 FORWARD AND LOO ABLATIONS

We show additional results for Section 3.2 in Figures 10a-10b. We also plot the pass rate vs compute leave one out experiments from Section 4 in Figure 12.

## A.10 ARE THE DESIGN CHOICES WORTH IT?

In Section 3.2, certain design choices alter asymptotic performance, such as loss type (Figure 4b) and FP32 precision (Figure 4c). However, in our LOO experiments with **SCALERL** (Figure 5), these components appear less critical individually (last column in the figure). This raises the question of whether certain design choices can be safely left at their "default" values.

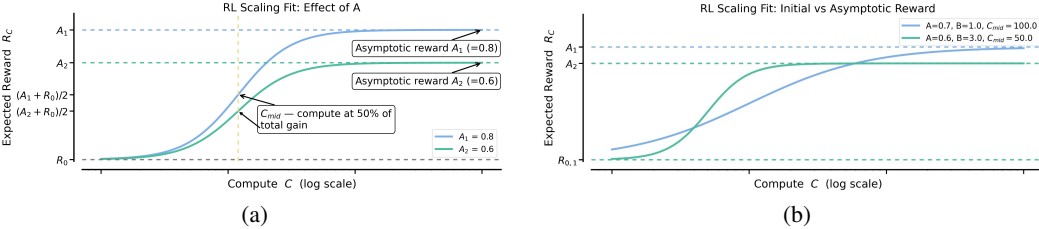

Figure 9: (a) Keeping all parameters same and only changing $A$. (b) A design choice can be less efficient yet reach a higher asymptote. When designing scalable methods, one should prioritize choices that raise the asymptotic ceiling $A$, since the ultimate goal is maximizing performance at scale.

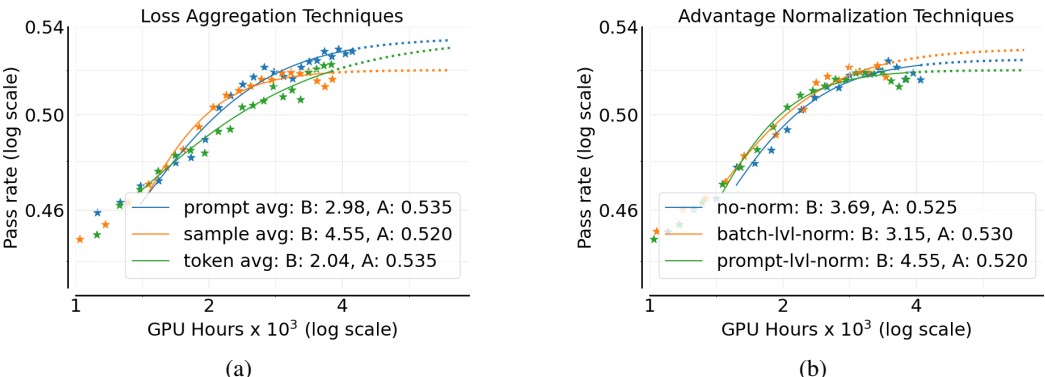

Figure 10: Comparing (a) loss aggregation, (b) different advantage normalization techniques.

We argue the answer is no. Even when a choice seems redundant in the combined recipe, it can still provide stability or robustness that can become decisive in other regimes. For example, while the FP32 precision fix makes little difference with dense 8B trained with **SCALERL** (Figure 5), it provides large gains in GRPO/DAPO-style losses by mitigating numerical instabilities. This indicates that its benefits extend beyond the specific **SCALERL** configuration we study. To further test this, we ran a leave-one-out experiment on the Scout 17Bx16 MoE and observed that FP32 precision improves overall scalability (Figure 7b).

A similar case arises with the loss type. In Figure 5, reverting to DAPO yields similar asymptotic performance to CISPO within **SCALERL**. Nonetheless, as we discuss in Appendix A.18, CISPO is markedly more robust to the choice of IS-clipping parameter $\epsilon_{\max}$, reducing the sensitivity of training to hyperparameter tuning. Moreover, it's also more efficient than DAPO, as seen in LOO experiment ($B = 2.01$ vs $B = 1.77$). This justifies preferring CISPO, even if a carefully tuned DAPO variant can perform similar asymptotically.

In summary, even when individual design choices appear redundant within the combined recipe, they often enhance training stability, robustness, or efficiency in ways that generalize across models and setups. **SCALERL** retains such components not just for marginal gains in a specific configuration, but because they address recurring sources of instability and variance that arise across reinforcement learning regimes.

## A.11 CONTROLLING GENERATION LENGTH

One common concern in reasoning RL is to control exploding generation lengths, which harms both training efficiency and stability (Appendix A.16). We consider two approaches: (a) *interruptions*, used in works like GLM-4.1V (GLM-V Team et al., 2025), and Qwen3 (Yang et al., 2025) and (b) *length penalties*, used in works like DAPO (Yu et al., 2025), Kimi (Kimi Team et al., 2025b), Magistral (Rastogi et al., 2025), and Minimax-M1 (MiniMax et al., 2025).

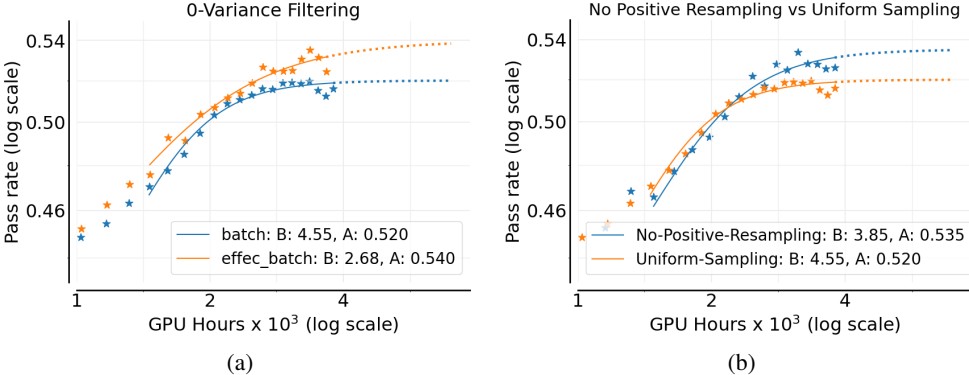

(a)  (b)

Figure 11: (a) **"Zero" variance filtering:** We filter out "zero" variance (accuracy 0 or 1) samples in a batch since they contribute zero policy gradient and find it achieves a higher asymptote, and (b) **Adaptive prompt sampling:** Filtering out prompts with pass rate $\geq 0.9$ in subsequent epochs results in a higher asymptotic performance.

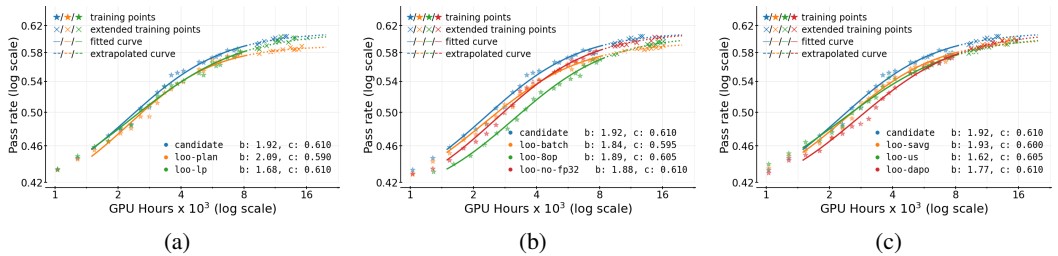

(a)  (b)  (c)

Figure 12: Comparison of different leave-one-out strategies using 16k GPU-hours budget. loo-plan refers to using prompt level advantage normalization, loo-lp means using length penalty, loo-batch refers to using the entire batch without any 0-variance prompts filtering. loo-8op refers using PPO-offpolicy-8, loo-fp32 means not using FP32 precision fix, loo-savg means using sample average loss aggregation, loo-dapo means using DAPO loss function instead of CISPO. Table in Figure 5 gives the values of $C_{min}$ in addition to $A$ and $B$. We notice that all methods have similar values of $A$ (within $\pm 0.02$ error margin range). Hence, all methods scale well, but affect efficiency parameters $B$ and $C_{mid}$.

**Interruptions** forcibly stop generation by appending a marker phrase such as "Okay, time is up. Let me stop thinking and formulate a final answer </think>", signaling the model to terminate its reasoning and produce a final answer. In our setup, the interruptions tokens are placed randomly in between $[10k, 12k]$ token length, to induce generalization to different generation lengths.

**Length penalties** instead reshape the reward. Following DAPO (Yu et al., 2025), we penalize overly long completions with a tolerance interval $L_{\text{cache}}$:

$$R_{\text{length}}(y) = clip\left(\frac{L_{\max} - |y|}{L_{\text{cache}}} - 1, -1, 0\right) \quad (9)$$

This penalty is added only to the correct traces, discouraging excessively long generations. In the length penalty experiment, we set $L_{\max} = 14k$ tokens and $L_{\text{cache}} = 2k$ tokens.

In Section 4, we compare length penalty and interruption at a scale of 16k GPU-Hours. We find that replacing interruption with length penalty in our final **SCALERL** recipe does not improve performance.

### A.12 PIPELINERL

Using the baseline setup, we ablated the off-policy parameter in PipelineRL (Figure 13). Both 4 and 8 off-policyness performed equally well, and we adopt 8 as the default setting when updating the baseline in Section 3.1.

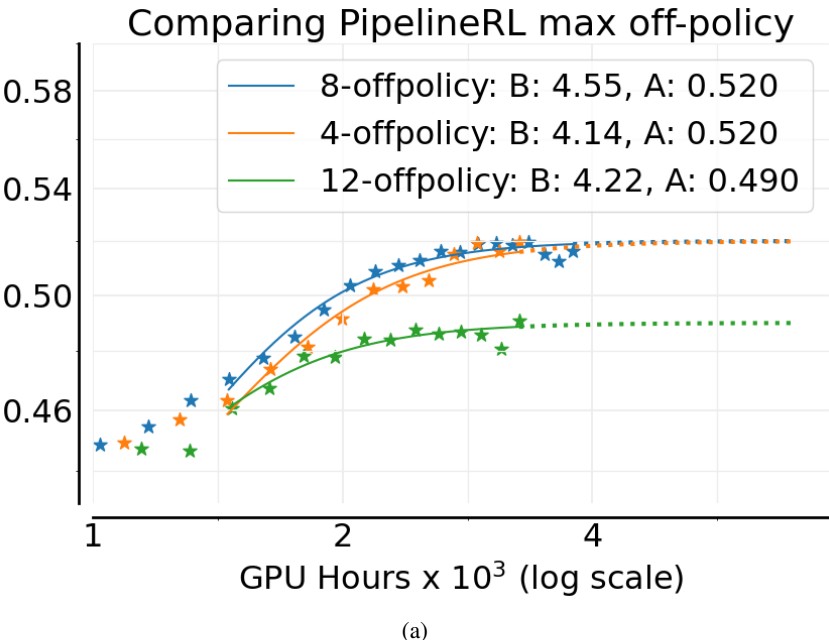

(a)

Figure 13: Different off-policy runs with PipelineRL

Why does PipelineRL consistently outperform the classic PPO-off-policy approach (Sections 3.1 and 4)? We attribute this to its closer alignment with on-policy training. In PPO-off-policy, generation and training proceed in alternating phases: the trainer operates strictly on batches that are as off-policy as the chosen parameter $k$, making updates based on stale rollouts. In contrast, PipelineRL operates in a streaming fashion. As soon as a batch is available, it is passed to the trainer; likewise, as soon as a model update is ready, it is shared back to the generators, who immediately use it—including in the continuation of partially generated traces. This tight feedback loop keeps training closer to the on-policy regime, reducing the mismatch between generator and trainer distributions.

Moreover, since there is not as much waiting in trainers and generators, it helps in training being more efficient as well.

### A.13 ENTROPY CURVES: SCALING BATCH SIZE

We tracked entropy on the held-out validation set throughout training. Across all experiments—spanning variations in batch size, number of tasks, generation length, and model scale—we observed a consistent overall decrease in entropy.

An interesting finding is that entropy may not always offer a predictive insight into the performance, as proposed by some recent works like Cui et al. (2025). In Section A.13, we plot entropy for **SCALERL** runs with batch sizes 768 and 2048. Despite the 2048-batch size run achieving much stronger downstream performance at every stage (Figure 17a), both runs followed nearly identical entropy trajectories per step (Section A.13). This highlights an important point - although entropy is sometimes used as a proxy for exploration, simply maintaining higher entropy does not translate into better generalization. Instead, larger batches reduced effective exploration similar to smaller batches, per step, yet still yielded substantially better performance - underscoring batch size as an important decisive factor.

Overall, our findings suggest that while entropy decreases consistently during training, it is not necessarily a reliable predictor of downstream performance. This observation reinforces the need to focus on algorithmic and scaling choices (e.g., batch size, off-policy method) in adddition to entropy dynamics when aiming for improved performance, both on training distribution as well as downstream task distribution.

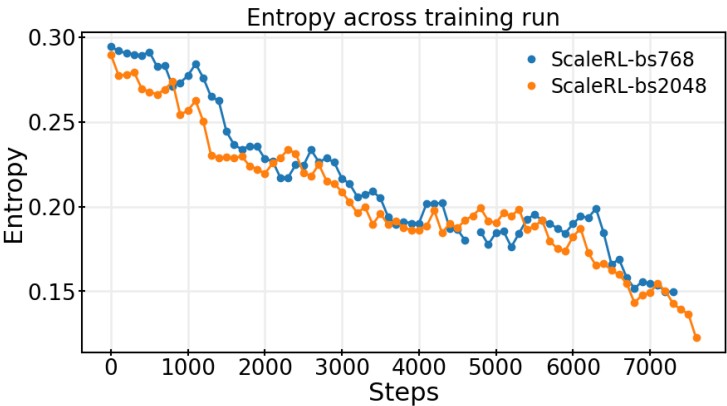

Figure 14: Comparing entropy of large and smaller batch size runs across training steps.

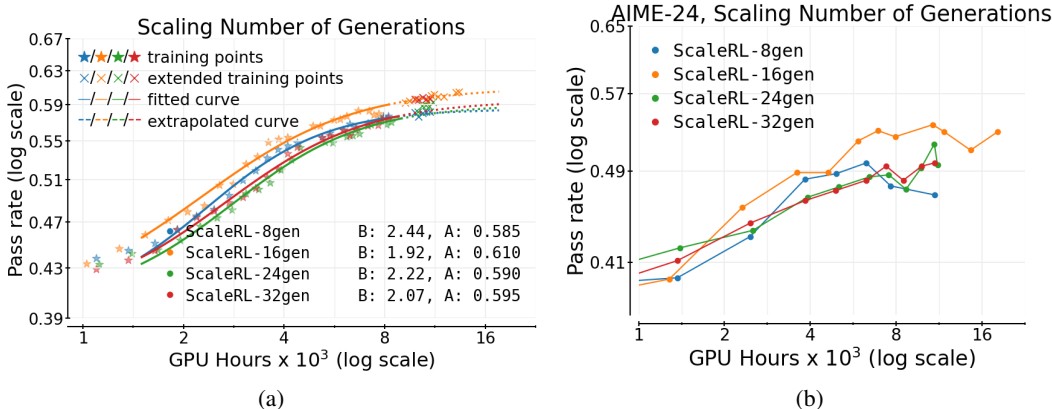

Figure 15: Scaling to (a) different number of generations per prompt, (b) Downstream performance of different number of generations per prompt

## A.14 SCALING ON MULTIPLE AXES

We provide the remaining scaling to different axes figure here in Figure 15 and Figure 16, and the corresponding downstream evaluation in Figure 18. We also provide the value of $A, B, C_{mid}$ in Table 1.

| Experiment | $C_{mid}$ | $B$ | $A$ |
|---|---|---|---|
| **SCALERL** | 2542 | 1.92 | 0.610 |
| **SCALERL**-32k | 11272 | 1.89 | 0.645 |
| **SCALERL**-8gen | 2542 | 2.44 | 0.585 |
| **SCALERL**-24gen | 3054 | 2.22 | 0.590 |
| **SCALERL**-32gen | 2936 | 2.07 | 0.595 |
| **SCALERL**-Scout | 4242 | 1.65 | 0.710 |
| **SCALERL**-bs512 | 2818 | 1.77 | 0.605 |
| **SCALERL**-bs2048 | 10909 | 1.70 | 0.645 |
| **SCALERL**-math+code, math curve | 2896 | 2.05 | 0.595 |
| **SCALERL**-math+code, code curve | 1675 | 1.09 | 0.615 |

Table 1: $C_{mid}, B,$ and $A$ values for the large scale runs in Section 5.

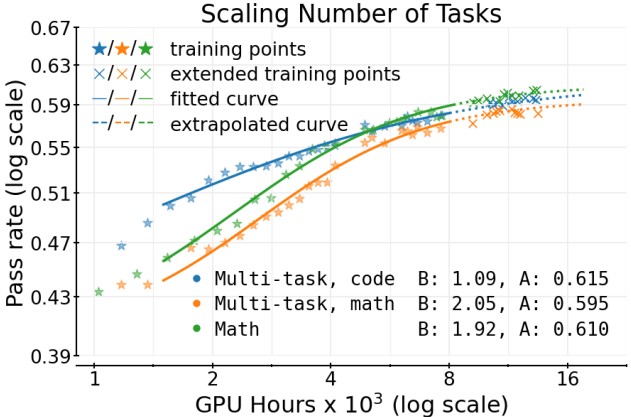

Figure 16: **SCALERL scales predictably on math and code**. We report both the code and math validation set performance on the joint math+code RL run; along with the math only **SCALERL** run as a reference. These results demonstrate that our sigmoidal compute–performance relationship holds across task mixtures, and that **SCALERL**'s scalability generalizes beyond a single domain training.

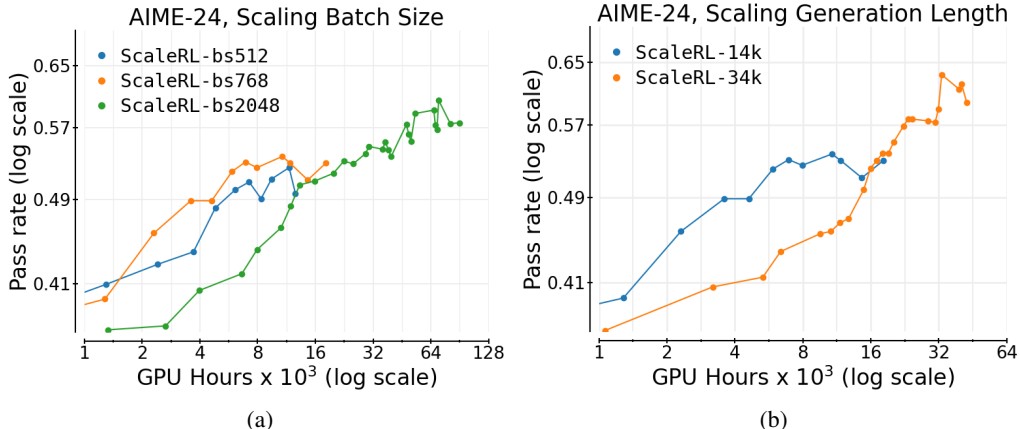

Figure 17: **Scaling RL batch size and generation length**. larger batch size and generation length are slower in training but settles at a higher asymptote. These show an inverse trend initially where smaller values seem better at lower compute budget, but reach a higher asymptotic performance at larger scale.

## A.15 DOWNSTREAM PERFORMANCE

In Figure 1, 15, 17, and 18, we report a representative set of downstream evaluation curves. These include **SCALERL** runs with batch sizes $\{512, 768, 2048\}$, long-context training run with 32k generation length, the large-model (Scout) training run, a multi-task run (math + code), and different number of generations per prompt (with fixed batch size) run. For each setting we plot performance against compute. Moreover, we see downstream performance better for experiments like larger batch sizes, longer generation length, and large model size - mirroring similar order for validation set curves.

## A.16 TRUNCATIONS AND TRAINING INSTABILITIES

Across our experiments we found that training instabilities were often linked to truncations. As generation length grew, many RL runs exhibited fluctuating truncation rates that sometimes increased over training. At batch size 768, we observed that truncations in the range of $10$–$15\%$ typically destabilized training, with performance degrading and not recovering without intervention. Exam-

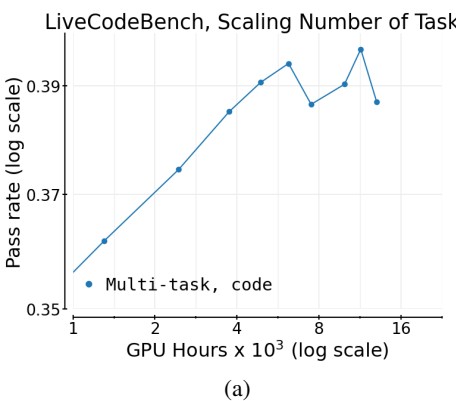 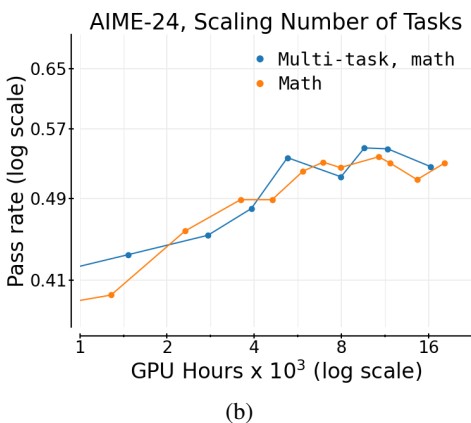

(a)                      (b)

Figure 18: Downstream performance of (a) different number of generations per prompt, on AIME, (b) LiveCodeBench (Jan-June 2025) performance on math+code run, (c) AIME-24 performance on math+code run

ples include the extended GRPO run in Figure 2, where instability correlated with rising truncation rates, and the updated baseline used in Section 3.2.

By contrast, **SCALERL** runs were more stable. On the 8B model, truncations remained below 5% for over 90% of training. At batch size 2048, truncations were slightly higher, occasionally approaching $\sim 7\%$. This increase was largely attributable to longer average generation lengths observed during training, which naturally raise the chance of exceeding the budget. Nevertheless, because the effective batch size (after excluding truncated samples) remained large, training stability was preserved. Intuitively, larger generation length budget should help reduce truncations. Training with 34k generation length (batch 768) remained stable - truncations briefly spiked to $\sim 4\%$ but quickly fell below 2%.

Larger models were even more robust. On the Scout run, truncations remained consistently below 2%, and for $> 90\%$ of training steps were under 1%. This likely reflects both the inherent ability of larger models to regulate generation length and their stronger instruction-following ability, which made interruption signals more effective.

Overall, we suggest practitioners monitor truncation rates closely. Our findings indicate that high truncation rates are a reliable warning signal of instability, while larger models, higher generation budgets, and careful design choices (as in **SCALERL**) substantially mitigate this risk.

### A.17 COMPARING PREVALENT METHODS

In Figure 2 we compared some popular training recipes with **SCALERL**. We briefly describe these existing recipes here.

**DeepSeek (GRPO)** This recipe mostly follows the DeepSeek Guo et al. (2025) work. We use GRPO as the loss function (Section A.2) with $\epsilon_{min} = \epsilon_{max} = 0.2$, sample average loss aggregation, and PPO-offpolicy-8 algorithm. We saw the training became unstable post 6k GPU Hours due to truncations (Section A.16).

**Qwen2.5 (DAPO)** This recipe follows DAPO Yu et al. (2025). It includes the DAPO loss function (Appendix A.2) with $\epsilon_{min} = 0.2, \epsilon_{max} = 0.26$ (Appendix A.18.1). This recipe uses PPO-offpolicy-8, and prompt average loss aggregation. The only change from the original DAPO paper (Yu et al., 2025) was regarding dynamically filling in the batch. Specifically DAPO drops 0-variance prompts and samples more prompts until the batch is full. In our codebase, this was not efficent because for PPO-offpolicy algorithm, we had generators pre-decide that each generator will generate rollouts for $\#\text{prompts}/\#\text{generators}$. Therefore, if a specific generator had more 0-variance prompts, it sampled further prompts to complete its share of $\#\text{prompts}/\#\text{generators}$. This could lead to other generators being

stalled and an overall slowdown. Hence, to get around this issue, we rather kept a larger batch size of 1280 (80 prompts, 16 generations each), and dropped 0-variance prompts from the batch. We noted that post-dropping, the effective batch was still greater than 768, what we used for **SCALERL**. Therefore, if at all, we gave some advantage to the DAPO recipe.

**Magistral**   This refers to the recipe used in Rastogi et al. (2025). It includes similar recipe as DAPO with the main difference being PipelineRL used as the off-policy algorithm.

**MiniMax**   This refers to the recipe used in MiniMax et al. (2025). It uses CISPO loss, FP32 precision fix at the LM head, PPO-offpolicy algorithm, and prompt average. Similar to DAPO, it drops 0-variance prompts as well and hence we give it a larger batch size of 1280 as well.

### A.18   LOSS TYPE - STABILITY AND ROBUSTNESS

As discussed below, GRPO/DAPO-style losses are highly sensitive to the choice of clipping ratio hyperparameter $\epsilon_{\max}$. In contrast, CISPO and GSPO show far greater robustness. For example, in Appendix A.18.2, varying $\epsilon_{\max}$ for CISPO between $\{4, 5, 8\}$ produced no significant differences in performance. For GSPO, the $10^{-4}$ clipping scale used in the original paper (Zheng et al., 2025a) did not work well in our setting. We therefore ablated across broader scales and found that once the correct order of magnitude was identified (e.g., $4 \times 10^{-3}$ and higher), performance was stable and largely insensitive to fine-grained changes (e.g., $\{4 \times 10^{-3}, 5 \times 10^{-3}\}$).

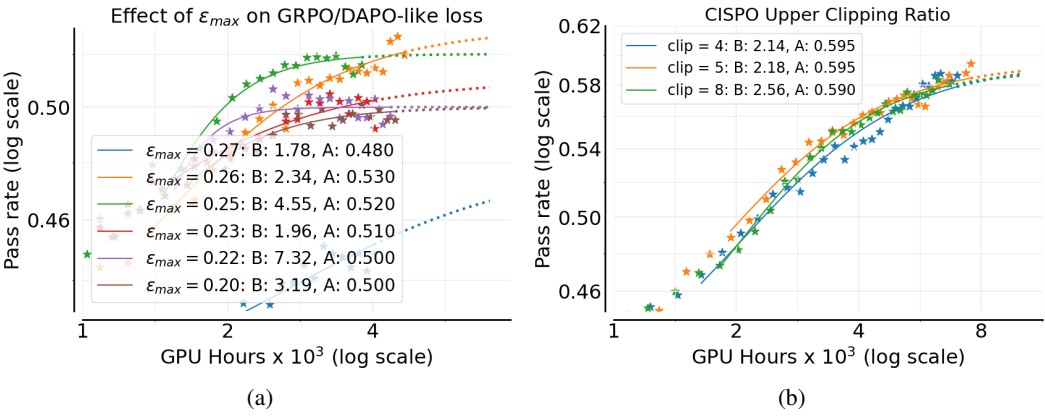

Figure 19: (a)Comparing upper clipping ratio of DAPO loss function. Change of $\epsilon_{max}$ fundamentally changes the asymptotic performance value $A$. (b) CISPO clipping ratio ablations

### A.18.1   DAPO CLIPPING RATIOS

In this section, we analyze the role of the clipping threshold $\epsilon_{\max}$ in DAPO Loss Function (eq. (8)). The hyper-parameter sensitivity of $\epsilon_{max}$ has been observed in prior work, for example, GRPO typically sets $\epsilon_{\max} = 0.2$, while DAPO uses $0.28$. However, beyond tuning sensitivity, we find that $\epsilon_{\max}$ directly alters the scaling behavior of the algorithm. As $\epsilon_{\max}$ increases, the terminal reward $A$ increases until an optimal range is reached, after which $A$ decreases again. This is a striking effect: unlike many hyper-parameters that merely shifts the convergence speed, $\epsilon_{\max}$ governs the asymptotic error itself.

### A.18.2   CISPO CLIPPING RATIOS

We ablate the higher clipping ratio for CISPO, keeping the lower clipping ratio fixed at $0$ (Figure 19b). Across a wide range of values, we find little difference in performance, indicating that CISPO is largely insensitive to this hyperparameter. This robustness mirrors our findings for GSPO (Section A.18.3), and stands in contrast to DAPO/GRPO-style objectives, which are highly sensitive to the exact choice of clipping threshold. Such stability under hyperparameter variation makes CISPO a strong candidate for default use in large-scale training.

### A.18.3 GSPO ABLATIONS

We ablate the clipping-ratio scale used in GSPO, as shown in Figure 20a. The default $10^{-4}$ scale as given in the GSPO paper Zheng et al. (2025a) does not scale the best for our 8B model. The $10^{-3}$ scale performs as well as, or better than, alternatives (Figure 20a) Given this scale, we further varied the upper clipping ratio in $\{4 \times 10^{-3}, 5 \times 10^{-3}\}$ and found $\{5 \times 10^{-3}\}$ yielded slightly better fit (Figure 20b).

An important observation is that GSPO is quite robust to the choice of clipping ratio. Once the correct scale is identified, most nearby values or even larger scale perform similarly. This robustness contrasts sharply with DAPO-style losses, which are highly sensitive to the exact value of the higher clipping ratio, as noted in Section 3.2.

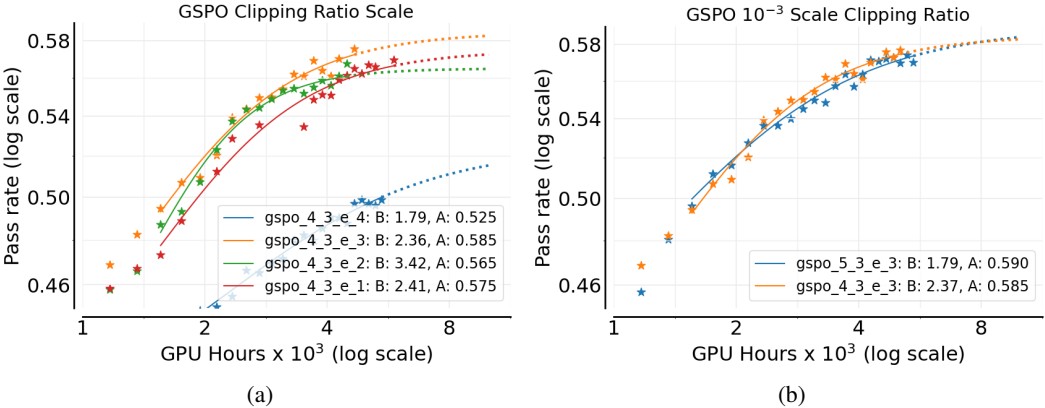

(a)             (b)

Figure 20: (a) GSPO Scale comparison. gspo_x_y_e_z in the legend means an upper and lower threshold of $\{x \times 10^{-z}$ and $y \times 10^{-z}\}$ respectively. (b) With $10^{-3}$ scale, we found similar performance for both 4_3_e_3 and 5_3_e_3, with latter performing slightly better.

### A.18.4 GSPO VS CISPO

Despite hyperparameter robustness, we encountered stability issues with GSPO. On multiple occasions, GSPO runs diverged mid-training, leading to sudden drops in performance. For 8B models, restarting from a stable checkpoint allowed recovery, but this strategy failed on larger models such as Scout, where instability persisted despite repeated resetting to a stable checkpoint. While we checked to the best of our ability for any implementation bugs, we did not find one.

Overall, while all three loss families can be competitive under tuned settings, CISPO offers the best balance of stability and robustness to hyperparameters, making it our recommended choice.

### A.19 ANALYZING FP32 PRECISION FIX

In Section 3.2 we introduced using FP32 at the LM Head as a way to decrease trainer-generator numerical mismatch. Here, we provide some additional experiments and analysis to support how this fix affects the training. First, we notice in Figure 21 that using the FP32 fix, the importance sampling gets more accurate (closer to 1, and hence log of it close to 0) at the 0th step of the training. We also provide a scatter plot in Figure 22 as an alternate way to visualize this. Second, in Figure 23, we log the clipping fraction, and the max and min value of importance sampling ratio during the training run of baseline with and without FP32 precision fix, and also the effect of using the fix on only either trainer or generator. We notice that while having FP32 precision fix on either the trainer or generator gives some benefit, having on both gives the most accurate IS ratios.

### A.20 SAMPLE EFFICIENCY AND SYSTEM THROUGHPUT

We analyze some of our ablations to understand the gains got from sample efficiency vs system throughput. First, in Figure 24, we plot the tokens processed per step by ScaleRL using PipelineRL,

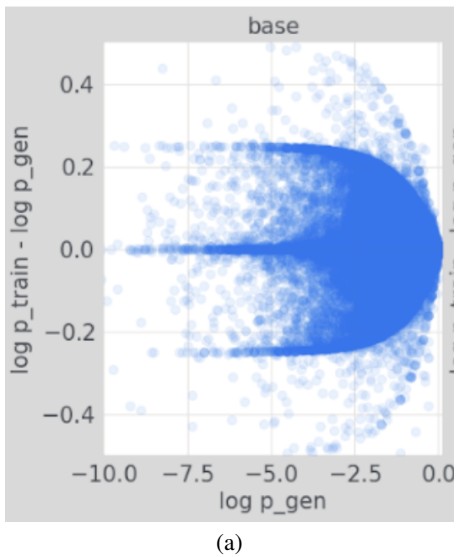 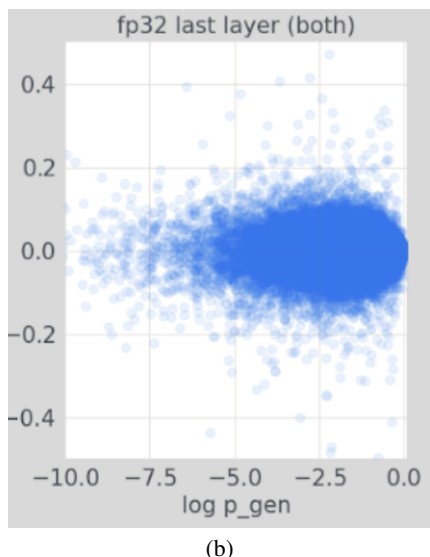

(a)                 (b)

Figure 21: FP32 fix at the LM Head reduces the mismatch in probabilities of trainers and generators. We plot the log of importance sampling vs log probability of the generator in (a) without the FP32 fix, (b) with the FP32 fix. Note that the above figures are on the 0th step of training, hence should ideally have a values of 0. Experiment conducted on the 8B Dense model.

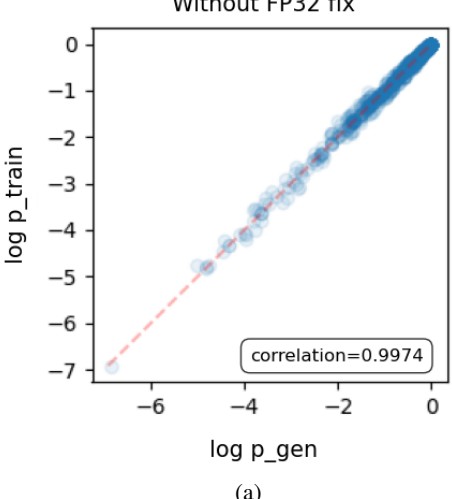 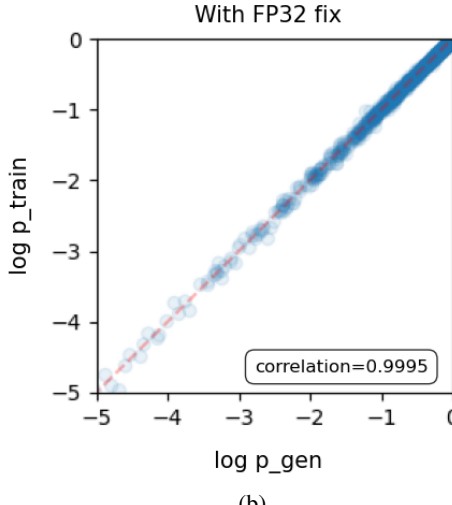

(a)                 (b)

Figure 22: Scatter plot of lof og probabilities with and without using FP32 precision fix. This logged at step=0, hence the expected behavior is to have all the points on x=y line. We notice that with the fix, values are much more aligned, with a higher correlation value as well.

vs LOO-PPO-Off-Policy-K. We notice that the sample efficiency of both the methods are similar, and the performance gains per step is similar as well. Although, we noticed in Figure 12b (candidate vs loo_8op) that PipelineRL is more efficient (has a larger value of efficiency parameter "b"). This means that most of the gains come from system throughput, and not an inherent sample efficiency.

The other two interesting comparisons where we expect sample efficiency to be affected are: (1) **Using No Positive Resampling**. Here, since we filter out the easy prompts, one may expect generations to be longer on the batch of no-so-easy prompts, and hence more tokens processed. But

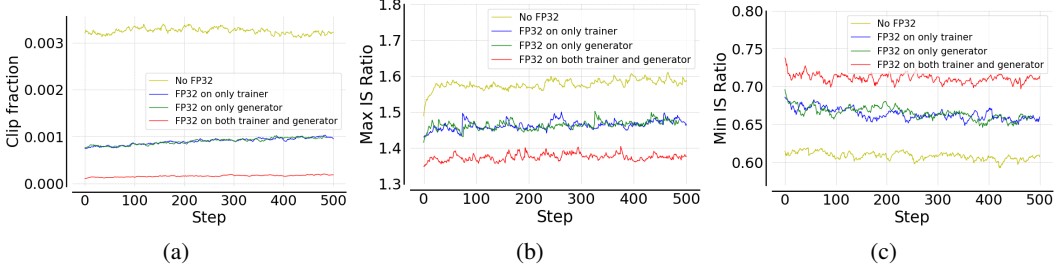

(a)  (b)  (c)

Figure 23: We compare not having FP32 fix, having FP32 fix on only the trainer, the generator, and on both trainer and generator. (a) The fraction of tokens clipped, (b) Maximum IS ratio in the batch, (c) Minimum IS ratio in the batch. We notice that having on either of just the trainer or generator makes IS ratio more accurate than having on none, but still leads to less accurate IS values compred to having on both. Using FP32 precision fox both makes the ratio much more accurate, with the maximum and minimum values being closer to 1 and having the least clipping.

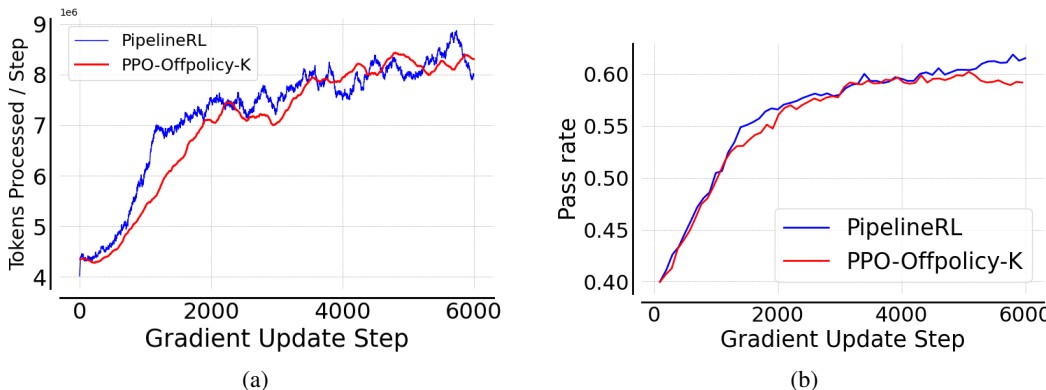

(a)  (b)

Figure 24: (a) We compare the tokens generated and processed per step by ScaleRL (which uses PipelineRL) and ScaleRL-LOO-PPO-Offpolicy-K. The latter means PipelineRL replaced with PPO-Offpolicy-K, while rest of the recipe being same as ScaleRL. We notice that the #tokens processed per gradient update step is approximately same for both the methods. (b) We also plot per step performance of these two methods as well, and observe similar per step gains, with PipelineRL performing slightly better. From these two figures, we deduce that the efficiency benefits of PipelineRL comes mainly because of system level improvements and not inherent sample efficiency

we notice in Figure 25a that's not necessarily the case, and the tokens processed is roughly same as if one uses uniform sampling (2) **Model Size**. As we increase model size, we expect the average generation length to be smaller for a given prompt. This is because a better model may require less thinking to solve the prompt. We observe that this is indeed the case in Figure 25b.

## A.21 ADDITIONAL DOWNSTREAM EVALUATION PLOTS

We discussed in Section 7 and Appendix A.15 some of the downstream performances of ScaleRL on AIME-24. Here, we present two additional plots, evaluating ScaleRL on 8B Dense and 17Bx16 MoE model on MATH-500 and AIME-25 datasets. We notice that similar to validation curves (Figure 6), the downstream performance improves in the expected way - roughly following a similar sigmoidal curve.

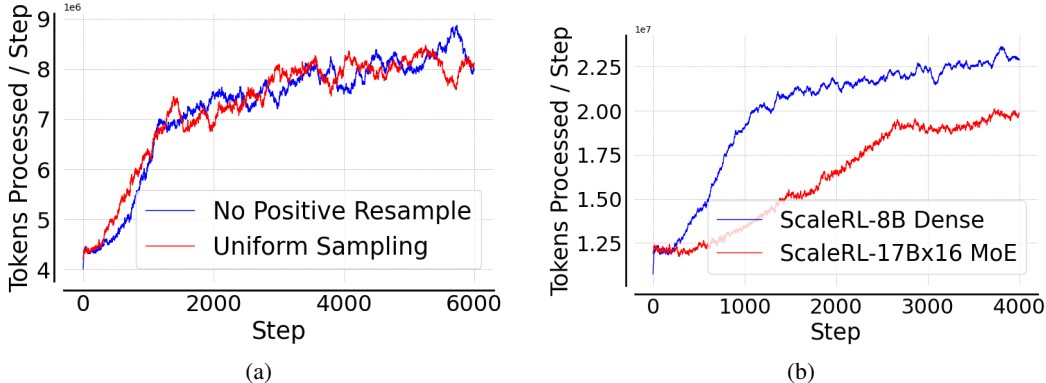

Figure 25: (a) We compare the tokens generated and processed per step by ScaleRL (which uses No Positive Resampling) and ScaleRL-LOO-Uniform-Sampling. The latter means No Positive Resampling replaced with Uniform Sampling, while rest of the recipe being same as ScaleRL. We notice that the #tokens processed per step is approximately same for both the methods. (b) We do similar analysis using ScaleRL but on different model sizes - 8B Dense and 17Bx16 MoE. Since a more capable model needs less thinking to solve a given prompt, we notice that it's more sample efficient.

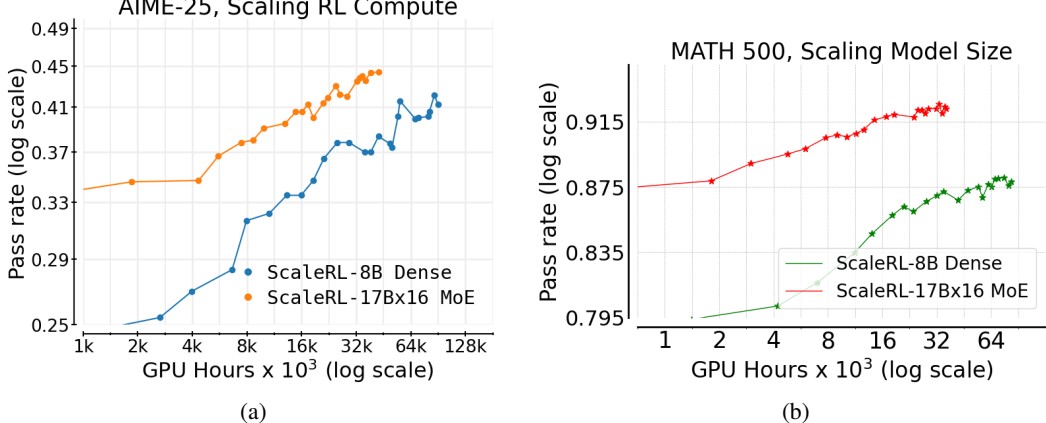

Figure 26: We evaluate ScaleRL on 8B Dense and 17Bx16 MoE, on (a) AIME-25 and (b) MATH-500 tasks. We notice that similar to AIME-24 (Figure 1), performance improves in the expected manner.

