# OpenReview forum: "The Art of Scaling Reinforcement Learning Compute for LLMs"
_ICLR.cc/2026/Conference — ICLR 2026 Oral_

### Official Review · Reviewer_abjx · 2025-10-22

**Soundness:** 3
**Presentation:** 3
**Contribution:** 3
**Rating:** 8
**Confidence:** 3

**Summary:**

The paper proposed a methodological breakthrough. The first systematic study of RL scaling laws. This paper aims to address a core pain point in the field of RL for LLMs: the lack of a scientific framework capable of predicting how RL performance scales with the investment of computational resources. Focus on the problem of unpredictability, research bottelneck on large-scale experiments and algorithm decision makeing difficulty. It show the system investigation of the RL post-training for LLMs. It introduces the concept of scaling laws from the pre-training domain into RL. Acknowledging that RL performance is bounded. The framework provides a new, quantitative language for evaluating and comparing different RL methods. ScaleRL itself is not a algorithmic invention. Instead, through rigorous empirical study, it combines the most effective existing components from the community. The authors propose a two-pronged solution, establish a predictive computation-performance fitting framework based on a sigmodial curve to model the relationship between reward and computation. Futuremore, propose a best-practice recipe scaleRL integrates a series of technical components.

**Strengths:**

1. Significant computational cost savings: The framework enables researchers to evaluate the potential of different RL methods using a small computational budget
2. Reduced risk in large-scale training: Using ScaleRL recipe, researcher can better predict the LLM training performance before large training.
3. The provided open-source training recipe on 100,000 GPU hours, which can achieves high performance ceiling, is valuable for community.

**Weaknesses:**

1. Lack of generalization research. The authors also mention it in paper. It hard to predict the generalization performance. Also the experiments focus mainly on math, it is hard to predict whether the findings can be generalized on more difficult, long horizon task, with different training data mixtures, under different architectures.
2. It doesn’t proposed the new RL algorithm, but a comprehensive large scale experiments on current existing  most effective algorithms.

**Questions:**

None.

---

> ### Author Response · Authors · 2025-11-28
> **Authors response**
>
> > Lack of generalization and multi-task training
>
> We provide preliminary results showing promise of our extension to multiple task setting, including math and code, in the Multi-task RL paragraph of Section 4, Appendix A.14, and Figure 14. Moreover, from our results of downstream tasks in (Figure 1, Figure 6, and “Generalization” paragraph in Section 6) showcase that ScaleRL indeed gives downstream performance improvements as well. Taking your feedback, we added further downstream evaluation results in Appendix A.18.3 of the revised draft, on AIME-24 and MATH-500. Moreover, we also added multi-task setting as an interesting future direction in the Future Work paragraph of Section 6 in our revised draft.
>
> > It doesn’t propose .. most effective algorithms.
>
> We address this point in our common rebuttal comment.
>
> ---
> We welcome further questions about the work, and if key issues are addressed, we would greatly appreciate an appropriate increase in score.

---

### Official Review · Reviewer_JrWQ · 2025-10-31

**Soundness:** 3
**Presentation:** 3
**Contribution:** 3
**Rating:** 6
**Confidence:** 4

**Summary:**

This paper studies how RL post‑training for LLMs scales with compute and proposes a practical recipe, ScaleRL, that is claimed to scale predictably and stably to very large budgets. The central methodological contribution is to fit a sigmoidal compute‑performance curve for pass rate (reward) vs. compute,. Using this framework, the paper ablates many design choices, assembles ScaleRL (PipelineRL with off‑policyness, CISPO loss with truncated IS, prompt‑level loss averaging, batch‑level advantage normalization, FP32 logits, zero‑variance prompt filtering, and "No‑Positive‑Resampling" curriculum), and claims that early‑compute fits extrapolate well to much larger runs. A major  result ofthe paper is that an 8B model trained for 100k GPU‑hours follows the curve predicted from the first 50k GPU‑hours, and that ScaleRL attains a higher fitted asymptote than several contemporary recipes. The paper further explores scaling along axes such as context length, batch size, and model scale (including a 17B×16 MoE), and reports downstream gains on AIME‑25.

**Strengths:**

I like that the paper reframes RL post‑training through a predictive, saturating scaling law for in‑distribution pass rate vs. compute, and leverages it to make early‑compute forecasts, which is an angle that is underexplored relative to pre‑training scaling laws. Also, the study spans ~400k GPU‑hours of ablations and 100k GPU‑hours long runs, which is unusually extensive for open academic RL studies in LLMs. Additionally, the predictive claim is backed by several "fit‑then‑extend" tests: e.g., fitting up to 8k hours and extrapolating to 16k for LOO runs, and fitting to 50k and matching 100k on the main 8B run. The main recipe of the paper, ScaleRL, seems to be a very stable and practical idea.

**Weaknesses:**

Limited task breadth and generalization evidence:

Most scaling fits are on verifiable math prompts drawn from Polaris‑53K; downstream evaluation focuses largely on AIME‑25. with some math+code multi‑tasking in the appendix. This leaves open how well the sigmoid fits and the ScaleRL recipe transfer to diverse RL setups (e.g., tool‑use, planning, safety‑constrained tasks, preference‑model rewards). A broader suite (beyond math/code) would strengthen the generality claim.

Compute as "GPU‑hours" conflates sample efficiency and throughput differences:

The x‑axis is GPU‑hours rather than effective tokens / updates / actor‑generated tokens. Because PipelineRL increases hardware utilization, improvements in "compute efficiency" may partially reflect systems throughput rather than intrinsic sample efficiency. Reporting both (hardware time and tokenized data processed) would make cross‑recipe comparisons more interpretable.

Misc. Notes:

I think that the Sigmoid choice is empirically motivated but not stress‑tested against alternatives. I see this as a necessary experiment.

The text sometimes describes “pass rate vs. log(compute)” but plots label compute in GPU‑hours on a log scale.

Some scope of the comparisons is missing. For example, value‑augmented baselines (e.g., VAPO/VC‑PPO‑style methods cited in App. A.1) are not included in the main scaling comparisons

**Questions:**

Can you report the effective tokens generated/consumed (actor and learner sides) for each method so we can decouple throughput from sample efficiency?

Fig. 3c shows a large asymptote jump from 0.52 to 0.61 with FP32 logits. Can you disentangle numerical‑mismatch reduction from any implicit regularization this might induce (e.g., via more accurate IS ratios)? Ablating only generator or only trainer in FP32 would help.

Did you compare sigmoid to Gompertz/Richards/saturating power‑law fits, and evaluate with AIC/BIC or hold‑out residuals?

---

> ### Author Response · Authors · 2025-11-28
> **Authors response**
>
> > Most scaling fits are on verifiable math prompts drawn from Polaris‑53K; downstream evaluation focuses largely on AIME‑25. with some math+code multi‑tasking in the appendix. This leaves open how well the sigmoid fits and the ScaleRL recipe transfer to diverse RL setups (e.g., tool‑use, planning, safety‑constrained tasks, preference‑model rewards). A broader suite (beyond math/code) would strengthen the generality claim.
>
> We agree with the sentiment that testing the predictive RL behavior on agentic training, tool calls, would be great. However, given the significant computational cost of these experiments (over 400k cumulative GPU hours ), we prioritized depth over breadth to establish a rigorous framework within the most standard RLVR testbed: reasoning models on math and code. We mention extending our work to other axes of RL as a future work (Section 6) in our revised draft.
>
> > The x‑axis is GPU‑hours rather than effective tokens / updates / actor‑generated tokens. Because PipelineRL increases hardware utilization, improvements in "compute efficiency" may partially reflect systems throughput rather than intrinsic sample efficiency. Reporting both (hardware time and tokenized data processed) would make cross‑recipe comparisons more interpretable.
>
> - Taking your feedback into consideration, we have included plots in Appendix A.18.2 and Figure 18, 19 of our revised draft, where we show the tokens processed per step over training steps and pass rate over training steps, comparing PipelineRL and PPO-Offpolicy-K in LOO setting. As you mention, we can deduce from those figures that indeed most of the gains of PipelineRL comes from increased system throughput, since per-step behavior is similar but performance in GPU-Hours for PipelineRL is better (Figure 8 (b), 3 (a) ).
> - We chose GPU-hours as our primary x-axis because we view scalable RL as a holistic combination of both algorithmic sample efficiency and system-level throughput. Practical scaling is constrained by wall-clock time and hardware availability. We believe reporting the composite metric (GPU-hours) is the most faithful representation of the "cost-to-result" ratio.
>
> > Can you report the effective tokens generated/consumed (actor and learner sides) for each method so we can decouple throughput from sample efficiency?
>
> Yes, we report it in our revised draft in Appendix A.18.2, for PipelineRL vs PPO-Off-Policy-K in Figure 17, and other interesting ablation comparisons like No Positive Resampling and Model Size in Figure 18. We found that most of the efficiency gains in PipelineRL comes from system throughput. And that a larger model can be more sample efficient than a smaller model.
>
> > Fig. 3c shows a large asymptote jump from 0.52 to 0.61 with FP32 logits. Can you explain the numerical‑mismatch reduction from any implicit regularization this might induce (e.g., via more accurate IS ratios)? Ablating only generator or only trainer in FP32 would help.
>
> - We added a section discussing the above in Appendix A.18.1. We find that FP32 precision fix indeed makes IS ratio more accurate (Figure 15, 16). In Figure 17, we also show the maximum and minimum value of IS ratio in the batch through training, where we notice that FP32 precision fix makes the maximum and minimum value much closer to 1, compared to not having this fix.
> - Our aim is to reduce the mismatch due to trainer/generator implementation discrepancy (He et al., 2025). In other words, given the same model on both trainer and generators, the ideal goal is to have no difference in log probabilities of trainer and generators for a given sequence. If we keep FP32 on only one of the machines, it will still lead to mismatch. Hence, we activated it on both sides.
> - Taking your feedback, we provide some ablations on using FP32 fix for just the trainer or generator in Appendix A.18.1, Figure17. We found that using either of just the trainer or generator has a similar effect, which is better than not having on either, but worse than having on both. Due to limited compute credits right now, we could run trainer/generator only experiment for around 500 steps so far, which doesn’t give as much signal on the validation set yet and hence we skip those. That said, IS ratios give clear signal already and we report those.
>
> [He et al., 2025]: Defeating nondeterminism in llm inference.
>
> ---
> We welcome further questions about the work, and if key issues are addressed, we would greatly appreciate an appropriate increase in score.

---

### Official Review · Reviewer_Cv6x · 2025-11-01

**Soundness:** 4
**Presentation:** 4
**Contribution:** 4
**Rating:** 8
**Confidence:** 3

**Summary:**

In this paper, the authors provide a comprehensive study of the scaling laws for large-scale distributed RL training. Based on the data collected as a result of this study, they propose and analyze a sigmoidal model, ScaleRL, that reasonably interpolates the training data and, crucially, closely extrapolates to additional points to predict the expected performance given a larger training budget. The authors consider the impact of several design choices when implementing RL, including the loss function, loss aggregation, and fp32 precision for logits. An extensive ablation study illustrates the interplay between several design methods and differentiates between substantial versus marginal impact on the peak performance of the training model. The authors provide several plots that illustrate the impact of a large number of design decisions on the training dynamics and use this data as a means to justify the design decisions that define the ScaleRL recipe to yield the best result.

**Strengths:**

- This is one of the most extensive large-scale RL studies I've had the pleasure to read. In particular, the ablation study illustrated in Figure 4 really highlights the trade-offs the authors considered during the course of this study.
- Given the emphasis and impact of RL as a key component of modern LLM pre-training this study will be particularly useful for researchers attempting to refine foundational models.
- ScaleRL gives a clear baseline recipe for researchers to use when beginning the RL training process and sheds additional light on how to scale RL and the relationship between several tradeoffs.
- This study provides a useful analogy to the heavily studied scaling laws for pre-training.
- I appreciate the fundamental view that a better understanding of the interplay between existing RL training hyperparameters may be more insightful than proposing yet another loss function or base RL algorithm.
- Comparison with

**Weaknesses:**

- Because the focus of the paper revolves around the comprehensive study of the different training parameters to consider for RL training it does not propose any novel methods, other than the sigmoidal model produced as a result of the collected data.
- The RL experiments are mainly conducted on a single model with 8B parameters and the model may not fit as closely as the structure and number of parameters are changed. However, this is offset somewhat by the experiments conducted on the larger 17B Llama-4 model and illustrated in Figure 5(b).
- It is still quite difficult to accurately interpret the leave-one-out experiments because of the complex interplay that may exist between several design decisions acting jointly.

**Questions:**

- Out of curiosity, all the graphs start at 1K total GPU hours. Before that threshold, are the graphs uninformative or too similar to provide any interesting information?

---

> ### Author Response · Authors · 2025-11-28
> **Authors response**
>
> > The RL experiments are mainly conducted on a single model with 8B parameters and the model may not fit as closely as the structure and number of parameters are changed. However, this is offset somewhat by the experiments conducted on the larger 17B Llama-4 model and illustrated in Figure 5(b).
>
> Indeed, we found a dense 8B model as the sweet spot to test different algorithmic components and scaling knobs (e.g., batch size, reasoning length) at larger compute scales (e.g., 6x more than ProRL) while being efficient enough to spend this compute for rigorous ablations. Unfortunately, a larger model would have limited the number of experiments we could conduct.
>
> Generally we have seen 8B to be a safe testbed and results to translate to larger scale 17Bx16 MoE (Figure 1 of the updated draft, and Section 4). Since our work is the first step towards building scaling laws for RL, we hope future work would explore model size as a scaling axis.
>
> > It is still quite difficult to accurately interpret the leave-one-out experiments because of the complex interplay that may exist between several design decisions acting jointly.
>
> Leave-One-Our(LOO) methodology provides strong evidence for the local optimality of the proposed recipe. By removing components one by one, we verify that every specific choice in ScaleRL contributes positively to either the asymptotic performance ($A$) or compute efficiency ($B$). Ideally, one would need to test all possible combinations of design choices. But that is an exponentially large set, and hence computationally intractable. For example, taking just 8 axes and, say, 2 design choice option for each axes, one would need 2^8 runs each of around 16k GPU Hours to test the combination. Hence, we rather revert to using LOO strategy, which is also a standard practice for validating complex composite algorithms in the field, also employed in the seminal "Rainbow" paper (Hessel et al., 2017) [1]
>
> > Out of curiosity, all the graphs start at 1K total GPU hours. Before that threshold, are the graphs uninformative or too similar to provide any interesting information?
>
> Skipping the low-compute regime for fitting curves is a standard practice in pretraining, as the low-compute region is usually noisy and is uninformative. That said, we observed for our stable and scalable RL runs, like those in Section 4, including them still led to stable fits. But for not so stable runs, like those in Section 3.2, often plateau prematurely or deviate from the sigmoidal trend due to transient instabilities (Appendix 6, Appendix 16). Excluding this region allows the fit to focus on the mid-to-high compute range where saturation behavior is clearer and more consistent. For uniformity, we decided to skip the initial 1.5K GPU Hours for all our experiments.
>
> Taking your feedback, we have also included this discussion in Appendix A.8 of the updated draft. We hope this answers your question and are happy to discuss further!
>
> > Because the focus ... the collected data.
>
> We address this point in our common rebuttal comment.
>
> [1] Rainbow: Combining Improvements in Deep Reinforcement Learning. Hessel et al., 2017
>
> ---
>
> We welcome further questions about the work, and if key issues are addressed, we would greatly appreciate an appropriate increase in score.

---

### Official Review · Reviewer_zx5H · 2025-11-01

**Soundness:** 3
**Presentation:** 3
**Contribution:** 3
**Rating:** 8
**Confidence:** 4

**Summary:**

This paper establishes a scaling law for RLVR training by running experiments on various RL algorithms with different implementation practices. The scaling law categories two key perspectives, asymptomatic reward gain, which defines the upper bound of the training process, and compute efficiency, which quantifies the performance gain versus used compute in RL training. The scaling law allows for predicting performance after RL training up to 100k GPU hours based on 40k GPU hours. The authors also investigate several practices in RLVR training, including loss aggregation, loss function, advantage normalization, dynamic fitlering, FP32 for policy head. Combining the best practice, the authors provide a recipe named ScaleRL that outperforms baselines in terms of both final performance and training efficiency.

**Strengths:**

1. The proposed scaling law is shown through extensive experiments.
2. The ScaleRL recipe considers a list of different techniques, which would provide valuable insights to the community.
3. Section 4 on scaling training compute across different axes reveals an interesting conclusion that configurations that are less efficient under limited compute may achieve better results given a larger compute.

**Weaknesses:**

1. In Sec 4, the large scale experiment on model scale makes a comparison between an 8B dense model and a large 17Bx16 MoE model. It would be better to include model scale experiment on comparing model sizes among dense models,

**Questions:**

1. Is "400,000 GPU hours" in the intro a typo? According to Figure 1, datapoints up to 100k GPU hours are predicted based on datapoints under 40k GPU hours.
2.  How could the scaling law used in practice? In the LOO experiments, the authors fit the parameters A&B in two ways, one is to fit both parameters simultaneously, while the other one is to fit B with a fixed A. In practice, suppose we run two different trials with different configurations, how to compare the two parameters if both A&B are fitted? Also, if we only fit B with a fixed A, how to determine A before at least one complete trial has been completed?

---

> ### Author Response · Authors · 2025-11-28
> **Author response**
>
> > Is "400,000 GPU hours" in the intro a typo? According to Figure 1, datapoints up to 100k GPU hours are predicted based on datapoints under 40k GPU hours.
>
> 400k GPU hours represents the cumulative compute GB200 GPU hours consumed by our study, including the 100k hour run, ablations in Section 3, the Leave-One-Out experiments in Section 4, RL scaling axes in Section 5, and the Appendix experiments. The 100,000 GPU-hours mentioned in Figure 1 refers to our single longest individual training run. We have updated our abstract and Figure-1 in the updated draft to clarify this.
>
> > How could the scaling law used in practice? In the LOO experiments, the authors fit the parameters A&B in two ways, one is to fit both parameters simultaneously, while the other one is to fit B with a fixed A. In practice, suppose we run two different trials with different configurations, how to compare the two parameters if both A&B are fitted? Also, if we only fit B with a fixed A, how to determine A before at least one complete trial has been completed?
>
> - Indeed, we describe our curve-fitting methodology in Appendix A.7. In practice, one should fit all three parameters ($A, B, C_{mid}$) simultaneously. We achieve this by performing a grid search over $A$ and $C_{mid}$, and fitting $B$ using standard non-linear least squares (e.g., scipy.optimize.curve_fit) for each candidate.
> - If two methods yield significantly different $A$ values, the one with the higher asymptote is generally preferred. However, if $A$ values are similar, fixing $A$ allows for a clearer comparison of compute efficiency ($B$).
>
> > In LOO experiments, the authors fit A&B in two ways, one is to fit both simultaneously, while the other one is to fit B with a fixed A
>
> Regarding the "fixed A" analysis in the LOO experiments (Figure 4):
> - First step (Standard Fit): We first fitted all three parameters ($A, B, C_{mid}$) for every run. These results are reported in the last column of the table in Figure 4.
> - Observation: We observed that the fitted asymptotic performance ($A$) for almost all LOO variants was very similar (mostly within a $\pm0.02$ margin, as noted in Appendix A.7).
> - Second step (Efficiency Analysis): Given that these methods reach a statistically similar asymptote, the differentiating factor becomes efficiency (how fast they reach that asymptote). To isolate and compare efficiency, we performed a post-hoc analysis where we fixed $A$ to the mean of the observed values ($0.605$) and re-fitted $B$. This allows for a direct comparison of the scaling exponent $B$.
>
> In practice, one should always fit $A$ and $B$ simultaneously first.
>
> > In Sec 4, the large scale experiment on model scale makes a comparison between an 8B dense model and a large 17Bx16 MoE model. It would be better to include model scale experiment on comparing model sizes among dense models,
>
> While a dense-to-dense comparison at scale would indeed be valuable, our primary goal within our available compute budget was to validate the generalizability of our “scaling” methodology across different architectures (MoEs and Dense), and to thoroughly study as many components of RL recipe as possible. That said, characterizing scaling behavior for dense models would be useful in building comprehensive RL scaling laws. We’d definitely try to include a dense model comparison in the final revision of our work, and expect 32B dense model to outperform 8B dense on compute matched comparison.
>
> ---
>
> We welcome further questions about the work, and if key concerns are addressed, we would greatly appreciate an appropriate increase in score.

---

### Author Response · Authors · 2025-11-28
**Thanks to Reviewers & Summary of Changes**

We thank the reviewers R1(zx5H), R2 (Cv6x), R3 (JrWQ), and R4 (abjx) for their feedback! All reviewers are in favor of acceptance and found the paper to be well-written and presented, with sound results and one of the most extensive large-scale RL studies using 400K GPU hours.

We first address a common concern raised by Reviewers R2 and R4 about **novelty** of our work: **The focus of our work is to contribute a scientific framework, rather than proposing yet another algorithmic heuristic.**
- Just as scaling laws revolutionized pre-training, we introduce a principled framework (sigmoidal laws) to predict RL performance, transforming the field from "art" into science.
- Novelty $\neq$ Scalability: We empirically show that recent "novel" algorithms (e.g., DAPO) often scale poorly compared to alternatives. Our derived recipe, ScaleRL, establishes a new state-of-the-art that outperforms these prevalent methods (Figure 2) in both asymptotic performance and compute efficiency.
- We solve a critical resource bottleneck by enabling the community to extrapolate large-scale performance from small-scale runs, a contribution which we believe is far more impactful than a marginal algorithmic variant.

To better showcase both our contributions, we have also updated *Figure 1*: (a) RL shows a predictive scaling behavior which can be studied using our framework, (b) ScaleRL - a stable and scalable recipe which can be scaled to up to 100k GPU Hours, and across model sizes, also giving downstream gains.


Based on reviewers’ comments, we have added a couple of discussion and analysis in Appendix A.18 ("Rebuttal"), which we summarize below:

  * Appendix A.18.1 (R3): **Further analysis of FP32** on the IS ratio through training.
  * Appendix A.18.2 (R3): **Results on sample efficiency and system throughput**, comparing methods like PipelineRL and PPO-Offpolicy-K, No Positive Resampling, and different model sizes.
  * Appendix A.18.3 (R4): **Additional downstream evaluation** of ScaleRL using both 8B and 17Bx16 MoE, on AIME24 and MATH-500.
* Appendix A.8 (R2): **Additional discussion on robustness of Scaling fits**

---

### Meta-Review · Area_Chair_kwMs · 2025-12-28

**Summary:**

The paper presents an extensive and timely systematic study of RL scaling laws for LLMs, consuming over 400,000 GPU-hours. It establishes a principled framework for RL scaling and analyze how various design choices, e.g., loss functions, aggregation, normalization, affect both asymptotic performance and compute efficiency. The paper has setup a important guideline for RL scaling and could be valuable to the community. All reviewers found the paper interesting and vote for acceptance.

**Reviewer Concerns:**

I think the following comments have been addresed
- Additional experiments regarding downstream evaluations, FP32, throughput, sample efficiency, and etc.
- Novelty of the work

Some of the concerns still persists
- Comparisons across model sizes.

**Reviewer Scores:**

I think JrWQ might raise the score to 8 because the authors have conducted a solid rebuttal and most of the concerns have been addressed.

---

### Decision · Program_Chairs · 2026-01-26

Accept (Oral)